# MITIGATING DATASET BIAS USING PER-SAMPLE GRADIENTS FROM A BIASED CLASSIFIER

## ABSTRACT

The performance of deep neural networks (DNNs) primarily depends on the configuration of the training set. Specifically, biased training sets can make the trained model have unintended prejudice, which causes severe errors in the inference. Although several studies have addressed biased training using human supervision, few studies have been conducted without human knowledge because biased information cannot be easily extracted without human involvement. This study proposes a simple method to remove prejudice from a biased model without additional information and reconstruct a balanced training set based on the biased training set. The novel training method consists of three steps: (1) training biased DNNs, (2) measuring the contribution to the prejudicial training and generating balanced data batches to prevent the prejudice, (3) training de-biased DNNs with the balanced data. We test the training method based on various synthetic and real-world biased sets and discuss how gradients can efficiently detect minority samples. The experiment demonstrates that the detection method based on the gradients helps erase prejudice, resulting in improved inference accuracy by up to 19.58% compared to the other state-of-the-art algorithm.

## 1 INTRODUCTION

Over the past decade, deep neural networks (DNNs) have been a focus of research owing to human-like performances in various tasks, e.g., image classification (Krizhevsky et al., 2012), object detection (Girshick, 2015), and image generation (Goodfellow et al., 2014). Despite these impressive results, deploying DNNs directly in real-world problems remains a significant challenge due to the difficulties in obtaining well-curated training sets. Specifically, unintendedly biased information in training sets causes prejudice, resulting in wrong decisions at inference time (Torralba & Efros, 2011; Shrestha et al., 2021). For instance, most "ski" images contain "skier." This unintended correlation can recommend a wrong shortcut for "ski" by examining the person.

Prior studies have employed human supervision (i.e., providing additional labels related to the bias) (Kim et al., 2019; McDuff et al., 2019; Singh et al., 2020) or giving a specific information about a biased domain (Lee et al., 2019; Geirhos et al., 2018), which is very expensive, to reduce the influence of a "skier" on "ski" class images. Recently, studies on replacing human labor with DNNs have been actively discussed (Li & Vasconcelos, 2019; Nam et al., 2020; Cadene et al., 2019; Bahng et al., 2020; Clark et al., 2019; Le Bras et al., 2020). Typically, two separate networks have been used: the biased network and the de-biased network. A biased model learns to replace human intervention and teaches a de-biased model using the prejudiced knowledge of the biased model.

Previous studies based on the two mentioned networks approaches have had two directions: *adjusting objectives* and *resampling*. Adjusting the objective refers to a method that obtains weighted loss or additional regularizers for each sample differently, where the weights or regularizers are obtained from the biased model (Nam et al., 2020; Cadene et al., 2019; Bahng et al., 2020; Clark et al., 2019). Methods that adjust the objective have been proposed frequently due to their several advantages, such as the implementational simplicity. However, according to (An et al., 2020), adjusting-objective approaches suffers instability with stochastic gradient descent (SGD) type optimizers owing to the high variance of learning weights per sample.

Alternatively, resampling methods could reconstruct balanced sets. Previous research (Le Bras et al., 2020; Li & Vasconcelos, 2019; Li et al., 2018b; Kim et al., 2021a) has focused on which samples

should be oversampled to build a balanced set. However, almost all prior works has focused on expensive human labor to tune hyperparameters related to the resampling densities of data points. For instance, it is difficult to set the amount of resampling of each sample without knowing how much biased the training set is.

The de-biasing methods, belonging to both adjusting-objective and resampling approaches are designed to emphasize samples that are hard for a model to learn. However, a few noisy labels (i.e., mislabeled samples) in the training set could significantly interfere with such de-biasing because it might be difficult to discriminate between two types of hard samples: noisy samples that the model should not learn and rare samples that have to learn. Although training sets have noisy labels in practice (Natarajan et al., 2013), to the best of our knowledge, no study considers the de-biasing problem under noisy label cases.

**Contribution.** In this study, we propose a score-based resampling scheme for de-biasing, which does not require hyperparameters related to training set reconstruction. Instead of hyperparameters, we construct two types of scores that leverage gradients of the biased model trained on the given biased set. The proposed scores determine the required proportions for each sample in training set reconstruction.

Before designing scores, we explain our hypothesis that *gradients have remarkable differences between samples generating prejudice and the others ,* and check it empirically. The hypothesis is based on the following observations. First, the gradient magnitudes (especially the Euclidian norm) of the samples which contributes to the prejudice are relatively smaller than those of the others. Second, samples that make prejudice and the others do not have significantly different gradient directions, measured by the estimated likelihood of the von Mises-Fisher (vMF) distribution (Banerjee et al., 2005; Lee et al., 2018).

Based on these observations, we propose two types of scores and a resampling-based de-biasing method. The proposed method consists of three steps similar to previous resampling studies (Li & Vasconcelos, 2019; Le Bras et al., 2020; Kim et al., 2021a). The first step is to train a biased model based on a biased raw training set. Afterward, the scores are obtained by the biased model. The mini-batch sampler generates balanced mini-batches using the scores, and it feeds them into an ultimate model for de-biased training.

In addition, we investigate the side effects of de-biasing methods when incorrectly labeled samples are in the training set. We observe significant performance degradations of de-biasing methods with a small portion of noisy data. To alleviate the negative side effects of noisy data, we first de-noise the training set using loss values from the our observation that noisy label samples have higher losses than clean samples and then run de-noising algorithms. The de-noising step successfully protects the de-biasing performance.

Finally, we demonstrate the effectiveness of our method using various biased benchmarks (Colored MNIST (Nam et al., 2020; Kim et al., 2019; Li & Vasconcelos, 2019; Bahng et al., 2020), Watermarked MNIST, Cartoon (Royer et al., 2020), CelebA (Liu et al., 2015), Biased action recognition (Nam et al., 2020), and ImageNet / ImageNet-A (Bahng et al., 2020; Deng et al., 2009; Hendrycks et al., 2019)). In most experiments, the proposed method outperforms the current state-of-the-art methods. In particular, the proposed method improves the accuracy of the unbiased test by $68.00\%$ from the baseline with a small performance degradation of the biased test of $0.91\%$. Moreover, we compare the proposed method with the previous resampling method, REPAIR (Li & Vasconcelos, 2019). The proposed method improves the average accuracy by $71.04\% \rightarrow 98.14\%$.

## 2 RELATED WORK

**Mitigating bias with human supervision.** In (Goyal et al., 2017; 2020), generated a de-biased dataset using human labor. A collection of studies (Alvi et al., 2018; Kim et al., 2019; McDuff et al., 2019; Singh et al., 2020; Teney et al., 2021; Ramaswamy et al., 2021; Tartaglione et al., 2021; Geirhos et al., 2018; Wang et al., 2018; Lee et al., 2019) aimed to mitigate bias based on one of two types of human supervision: *explicit bias labels* and *implicit bias information*. Other studies (Alvi et al., 2018; Kim et al., 2019; McDuff et al., 2019; Singh et al., 2020), have used bias labels for each sample to reduce the influence of the bias labels when classifying target labels. (Tartaglione et al.,

2021) proposed the EnD regularizer, which entangles target correlated features and disentangles biased features. Several authors (Alvi et al., 2018; Kim et al., 2019; Teney et al., 2021) have designed DNNs as a shared feature extractor and multiple classifiers. In contrast to the shared feature extractor methods, (McDuff et al., 2019; Ramaswamy et al., 2021) constructed a classifier and conditional generative adversarial networks, generating test samples to check whether the classifier is biased. (Singh et al., 2020) proposed a new overlap loss defined by a class activation map (CAM). The overlap loss reduces the overlapping parts of the CAM outputs of two bias labels and target labels. Different approaches (Geirhos et al., 2018; Wang et al., 2018; Lee et al., 2019) have assumed that the characteristics of bias features are known and prevent learning the implicit bias features. To avoid the known texture bias of ImageNet (Deng et al., 2009), (Geirhos et al., 2018) generated stylized ImageNet, and (Lee et al., 2019; Wang et al., 2018) inserted filter in front of the models so that the influence of the backgrounds and colors of the images can be removed.

**Mitigating bias without human supervision.** To reduce human intervention, recent research (Clark et al., 2019; Cadene et al., 2019; Bahng et al., 2020; Nam et al., 2020; Li & Vasconcelos, 2019; Li et al., 2018b; Le Bras et al., 2020; Darlow et al., 2020) has used ensemble-based methods, using two separate networks with different purposes. The *biased model* learns biased information and even empowers it. Then, the biased model is used to train the *de-biased model* as a pathfinder, which provides information about the bias features.

The ensemble idea has been used as a key component in several studies, (Clark et al., 2019; Cadene et al., 2019; Bahng et al., 2020; Nam et al., 2020). (Cadene et al., 2019) proposed a method known as RUBi, which multiplies the sigmoid output of the biased model by the softmax output of the de-biased model. (Clark et al., 2019) summed the output of the two models to train the de-biased model with an entropy regularizer. (Bahng et al., 2020) proposed ReBias algorithm using the Hilbert-Schmidt independence criterion (HSIC) as an objective of the de-biased model, by independent of the biased model. Learning from failure (LfF) (Nam et al., 2020) used generalized cross-entropy-based training that emphasizing the usual samples but ignoring unusual samples.

In contrast, biased and de-biased models are trained sequentially (Li & Vasconcelos, 2019; Li et al., 2018b; Le Bras et al., 2020; Darlow et al., 2020). Researchers computed the weights to reconstruct the training set from the biased model and trained the de-biased network on the reconstructed training set. The REPAIR (Li & Vasconcelos, 2019) and RESOUND (Li et al., 2018b) methods proposed mutual-information-based weights to create a downsampled dataset. In addition, AFLite (Le Bras et al., 2020) proposed accuracy-based weights from multiple biased models, unlike LAD (Darlow et al., 2020) and BiaSwap (Kim et al., 2021a), which generated new images using an autoencoder based on the latent representation of the biased model.

## 3 BIAS PROBLEM FOR TRAINING A CLASSIFIER

### 3.1 UNINTENDEDLY BIASED TRAINING

Suppose that a training set $D$ is composed of images $x$, as illustrated in Figure 1. Each image can be described by a set of attributes $\{a_1, ..., a_k, ...\}$, (e.g., {"digit 0", "digit 1",... "red", "green", ... , "thick",...}). The goal of the training classifier is to find a model $f_\theta$ that correctly predicts the $C$ target attributes "digits," $y = \{a_{t_1}, ..., a_{t_C}\}$ ={"digit

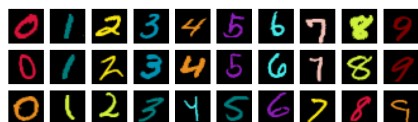

Figure 1: Colored MNIST.

0", "digit 1",...}. Remark that target attributes are also interpreted as *class*. However, we focus on the case in which another attributes exists, "colors," where $b = \{a_{b_1}, ..., a_{b_C}\} = $ {"red","green", ...} $\neq y$, which is highly correlated to the target "digits" (i.e., $H(a_{t_1}|a_{b_1}) \approx 0$). Such an correlated attributes and training set are denoted as the *bias attributes* and *biased training set*, respectively. In addition, the "color-digit" correlated and uncorrelated samples, in the top two rows and the bottom row in Figure 1, are called the *majority* and *minority*. We use the notations $M$ and $m$ for the majority and minority sets, respectively. For example, "red-digit 0" image is majority sample, while "orange-digit 0" one is minority sample.

When we train a model based on the biased training set $D$, the model may infer the bias attributes $b$, not target attributes $y$, which is problematic because the target is $y$, not $b$, and we refer to this as an

*unintended bias problem.* For example, if the model trained on the images in Figure 1 suffers from this problem, this model outputs 4 when *orange,*0 image is given.

## 3.2 WHEN DOES THIS PROBLEM HAPPEN AND INTENSIFY?

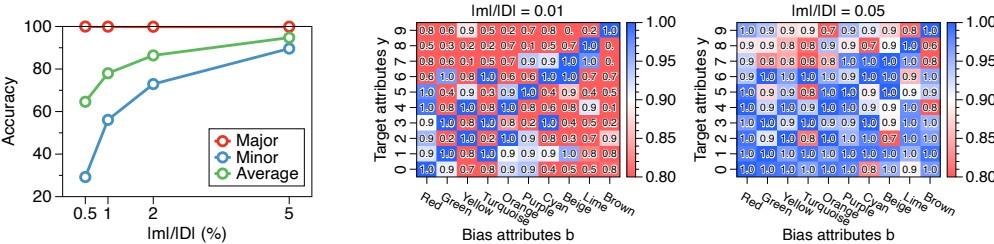

Figure 2: Accuracy vs. $|m|/|D|$.    Figure 3: Confusion matrices for $|M|/|D| = 99\%$ and $95\%$.

(Nam et al., 2020) argued that this problem arises when two conditions are met simultaneously: (C1) bias attributes $b$ is easier to learn than target $y$, and (C2) highly correlated $b$ to $y$. We focus on observing the effect of (C2) on the unintended bias problem among the two conditions when (C1) holds. To understand (C2) precisely, we check the effect of the minority ratio $|m|/|D|$ on the accuracy. A detailed description of the training setting is provided in Appendix A.2.

We investigate the test accuracy of the samples in the majority and minority sets. As in Figure 2, the average performance drop of the minority samples intensifies when the minority ratio $|m|/|D|$ decreases. On the other hand, the accuracy of the majority samples for all cases are not dropped. More precisely, we plot the confusion matrix to see how poor the performance is when the bias attributes $b$ and the target attributes $y$ are not aligned according to the minority ratio $|m|/|D|$. As in Figure 3, the increments of the minority ratio does not make the performance change of the aligned samples located in the diagonal, but other entries change a lot. These results can be interpreted as when the ratio $|m|/|D|$ becomes small, the model highly likely infer the bias attributes $b$ rather than target attributes $y$.

## 4    DE-BIASING USING PER-SAMPLE GRADIENT INFORMATION

When a given training set is biased, the model learns $b$ over the intended target $y$, and it is strengthened when $|m|/|D|$ approaches zero. If this is the case, it is desirable to modify the training set to a larger $|m|/|D|$ so that the model learns the target $y$ rather than the biased attribute $b$. To do so, we considered constructing a new training set $D' = \{(x'_i, y'_i)\}_{i=1}^N$ such that $|m'|/|D'|$ increases.

Similar to prior resampling-based work (Li & Vasconcelos, 2019), we sequentially train two separated networks: biased model $f_{\theta_b}$ and de-biased model $f_{\theta_d}$. The biased model, $f_{\theta_b}$ is trained based on the raw training set $D$. After sufficiently converging to point $\theta_b \to \hat{\theta}_b$, the model computes gradients $\nabla\theta_i = \nabla_\theta \mathcal{L}_{CE}(x_i, y_i; \theta)$ of all the samples $(x_i, y_i) \in D$. Then, we compute the sampling probability $(p_s(i))$ of the $i$th sample using the per-sample gradient $\nabla\theta_i$, to break (C2) by balancing between $M$ and $m$. Finally, we train the de-biased model $f_{\theta_d}$ using a rejection sampler that creates a balanced training set $D'$ based on the sampling probability $p_s(i)$.

### 4.1    PER-SAMPLE GRADIENT-BASED SCORES

As the first step of de-biasing, we train the biased model without modifying training set $D$. The converged model $f_{\hat{\theta}_b}(\cdot)$ contains in-depth information about the distribution of the training set in its model (especially, through the per-sample gradient). Therefore, we leverage the per-sample gradient, $\nabla\theta_i = \frac{\partial \mathcal{L}_{CE}(x,y;\hat{\theta}_b)}{\partial \theta}|_{x_i,y_i}$ of $(x_i, y_i)$, which is dissolved in the biased model.

**Why the per-sample gradient?** The majority and minority samples exhibit a clear difference in the per-sample gradient for the model $f_{\hat{\theta}_b}$ owing to the following reasons. (R1) Each sample has its own loss landscape, and similar representation samples have similar loss landscapes (see Figure 4(a)). In other words, the minority samples have different loss landscapes from those of the majorities, which implies that a local minimum cluster is composed of similar representation samples. (R2)

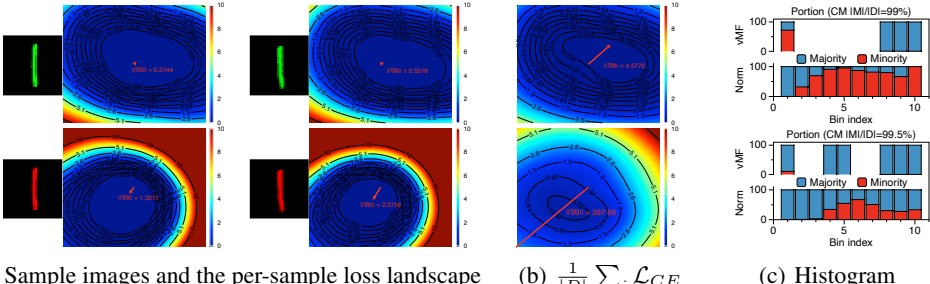

(a) Sample images and the per-sample loss landscape    (b) $\frac{1}{|D|}\sum_i \mathcal{L}_{CE}$    (c) Histogram

Figure 4: Loss landscapes of the model trained on the biased training set. (a) and (b) represents the per-sample and aggregated loss landscapes. Top and bottom row depict the results of the case. The red arrows are the convergence point and minimal point. We draw the loss landscapes based on $\|\nabla\theta\|$. The values are gradient direction and loss difference (Li et al., 2018a). See Appendix A.2 for the detailed experiment setting. (c) represents the ratio of majority and minority samples at each bin respect to the directional and magnitude.

From the averaging property of the batch-based optimization methods, the influence of the minority samples is diluted of according to their ratio. Thus, the model trained on the biased training set converges to a proximity point of the minimum point of the majorities and it is reflected in the gradient of each sample. The red arrows in Figure 4(a) and 4(b) indicate that **the majority samples (top) have shorter gradients than the minorities** (bottom) for both per-sample and aggregated cases. Similarly, **the direction of both cases is significantly different**. Additionally, as seen in Figure 4(c), the portion of minority samples in the lower directional likelihood and higher magnitude bins is much larger than the majority samples.

**Gradient-based score.** Based on the intuition of per-sample gradient and empirical results, we design a gradient-based score $S(i) = S\left((x_i, y_i); \hat{\theta}_b\right)$, which is computed for each class separately. The score is computed using two base scores, the magnitude score $M(i)$ and direction score $D(i)$, which are defined as follows:

$$M(i) = \left(\frac{(1/\|\nabla\theta_i\|_2)}{\sum_{j=1}^{N}(1/\|\nabla\theta_j\|_2)}\right), \quad D(i) = \left(\frac{p_{\mathrm{dir}}(i)}{\sum_{j=1}^{N} p_{\mathrm{dir}}(j)}\right). \tag{1}$$

(1) Magnitude score $M(i)$: The contribution to prejudicial training is inversely proportional to the magnitude of its gradient ($\propto 1/\|\nabla\theta_i\|$). We used Euclidian distance to measure the magnitude.

(2) Direction score $D(i)$: The contribution to prejudicial training is also proportional to the directional likelihood ($\propto p_{\mathrm{dir}}(i)$). For the directional likelihood, the estimated von Mises-Fisher (vMF) distribution is used to understand the concentration of the directional distribution. The vMF distribution is defined as follows:

**Definition 4.1** (von Mises-Fisher Distribution). The pdf of the vMF$(u, \kappa)$ is given by

$$f_d(v; u, \kappa) = C_d(\kappa)e^{(\kappa u^\top v)},$$

on the hypersphere $S^{d-1} \subset \mathbb{R}^d$, where $\kappa$ determines how much samples are concentrated based on the distribution along the mean direction $u$, and $C_d(\kappa)$ is a normalization constant determined by dimension $d$ and the concentration parameter $\kappa$.

To estimates the parameters of the assumed vMF distribution, we use approximated maximum likelihood estimates (MLE) solutions $\hat{u}$ and $\hat{\kappa}$. For normalized gradient vectors $\overline{\nabla\theta_i} = \frac{\nabla\theta_i}{\|\nabla\theta_i\|}$, the MLE solutions $\hat{u}$ and $\hat{\kappa}$ can be obtained from $\hat{u} = \frac{\sum_{i=1}^{N}\overline{\nabla\theta_i}}{\|\sum_{i=1}^{N}\overline{\nabla\theta_i}\|}$ and $\hat{\kappa} \approx \frac{\bar{r}(d - \bar{r}^2)}{(1 - \bar{r}^2)}$, where $\bar{r} = \frac{\|\sum_{i=1}^{N}\overline{\nabla\theta_i}\|}{N}$ (see Appendix G).

Then, the normalized directional score $D(i)$ is defined in (1), where $p_{\mathrm{dir}}(i) = f_d(\overline{\nabla\theta_i}; \hat{u}, \hat{\kappa})$ is the approximated vMF distribution.

To relax the computational constraint, we select $k$-dominant sub-dimensions, denoted by the set $K = \underset{K \subset N, |K|=k}{\arg\max} \sum_{j \in K}\left(\sum_i |\nabla\theta^j|_i\right)$, where $\nabla\theta^j$ corresponds to the gradient of the $j$-th dimension.

### 4.2 DE-BIASING USING GRADIENT-BASED SCORES

**Balanced batch selection.** By the definitions of the scores, the sample in the majority set $M$ (or the minority set $m$) must have a higher (or smaller) $M(i)$ and $D(i)$, respectively. Each sample must interact inversely with the two scores to treat the samples in the minority set $m$ similarly to the samples in the majority set. To do so, we design aggregated score $S(i)$ based on the harmonic mean, and sampling probability $p_s(i)$ as follows:

$$S(i) = \left( \frac{\lambda}{M(i)} + \frac{1}{D(i)} \right)^{-1}, \quad p_s(i) = \frac{C}{S(i)},$$

where $\lambda$ is the balancing hyperparameter to emphasize a specific score between $M(i)$ and $D(i)$, and $C$ is the normalization constant defined as $C = \min_i S(i)$, which makes $p_s(i) = 1$ for the smallest $S(i)$ sample. From the harmonic-mean, $S(i)$ has a higher value when one of the scores is high.

To construct a balanced mini-batch, our algorithm use a rejection-sampling-based uniform sampler, Algorithm 1. Each sample $i$ is rejected with a probability $1 - p_s(i)$, which implies that the rejection is proportionally to the scores. From the property of rejection sampling, such a sampler generates uniform batches.

---

**Algorithm 1** Uniform Sampler
1: Sampling prob. $p_s$, Batch size $k$
2: **while** $|B| < k$ **do**
3:     $i \sim \mathcal{U}(1, N), p \sim \mathcal{U}(0, 1)$
4:     **if** $p \leq p_s(i)$ **then**; $B = B \cup i$
5:     **end if**
6: **end while**

---

**Gradient-based de-biasing.** The ultimate de-biasing algorithm consists of three steps. First, the biased model $f_{\theta_b}$ is trained on the raw data samples with GCE loss (Zhang & Sabuncu, 2018; Nam et al., 2020). The GCE loss controlled by $q$ guides the biased model to ignore the minority samples in the training set. Then, all raw samples are input into the biased model to compute $p_s(i)$. The last step is to train the de-biased model $f_{\theta_d}$ using a balanced mini-batch sampler. This sampler generates balanced mini-batch based on the sampling probability $p_s(i)$.

---

**Algorithm 2** De-biasing using the uniform sampler
1: $D, \lambda, q$, lr $\eta, T, k$, aug. alg. $\mathcal{A}(\cdot)$, sampling alg $\mathcal{S}(\cdot)$
2: **/** ✱✱ **STEP 1: Train** $f_{\theta_b}$ ✱✱**/**
3: **for** t=1,2,...,T **do**
4:     Draw a mini-batch $(X, Y) = \{(x, y)\}_{i=1}^{k}$ from $D$
5:     Update $\theta_b \leftarrow \theta_b - \frac{\eta}{|D|} \nabla_\theta \sum_{(X,Y)} \mathcal{L}_{\text{GCE}}(x, y; q)$
6: **end for**
7: **/** ✱✱ **STEP 2: Calculate** $p_s(i)$ ✱✱**/**
8: Calculate $p_s(i)$ for all $i \in D$
9: **/** ✱✱ **STEP 3: Train** $f_{\theta_d}$ ✱✱**/**
10: **for** R **do** t=1,2,...,T
11:     Draw a mini-batch $\{(x, y)\}_{i=1}^{k}$ from the Alg. 1: $\mathcal{S}(p_s(i))$
12:     Update $\theta_d \leftarrow \theta_d - \frac{\eta}{|D|} \nabla_\theta \sum_{(X,Y)} \mathcal{L}_{\text{CE}}(\mathcal{A}(x), y)$
13: **end for**

---

We use the typical image data augmentation method $\mathcal{A}(\cdot)$ (e.g., rotate, resize) to avoid over-fitting.

## 5 DE-NOISING A SMALL PORTION OF THE NOISY LABELS

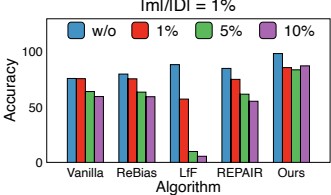

Figure 5: Acc. drops.

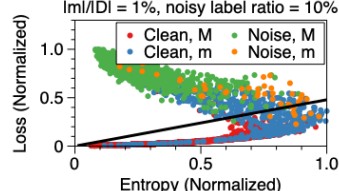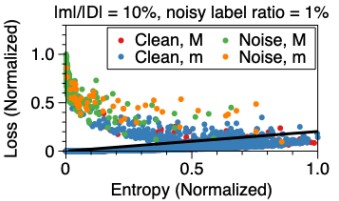

Figure 6: Entropy vs Loss.

**Why is noisy label a problem for de-biasing algorithms?** All de-biasing algorithms work by emphasizing rare samples. For example, the proposed method operates by reconstructing mini-batches in proportion to the rarity. However, when a small portion of incorrectly labeled samples, called *noisy labels*, is in the training set, the de-biasing algorithms accentuate the noise labels due to their mechanism of enlarging the influence of the rare samples. Therefore, as presented in Figure 5, all de-biasing algorithms suffer a performance decline.

**Distinguishing minority and noisy labels via compensated loss.** Recent de-noising methods (Yu et al., 2019; Yi & Wu, 2019; Li et al., 2019; Kim et al., 2021b) have mainly focused on ignoring unusual sample, which may be noisy labels. However, by comparing de-noising mechanism with (Kim et al., 2021b), previous de-noising algorithms also ignores minority samples which must be highlighted not discarded, see Appendix I for details. Therefore, we design a module to delete noisylabels in which the minority samples are preserved. To do so, we observe the loss statistics of the minority samples and noisy labels under various Colored MNIST settings with

$(|m|/|D|,$ noisy label ratio$) = (0.01, 0.1)$ and $(0.1, 0.1)$ (see Appendix B.3). As depicted in Figure 6, noisy labels have a higher loss than clean labels, regardless of whether the sample is in the majority or the minority set. The trained model that ignores rare samples, outputs the correct results, whereas the given label for calculating the loss value is incorrect.

However, the loss of the clean minority samples increases, owing to the rarity. To compensate, we suggest the compensated loss $\bar{\mathcal{L}}_i$, reducing the influence of the rarity, and the splitting criteria to distinguish between the clean samples and noisy labels:

$$D_{\text{clean}} = \left\{ (x_i, y_i) | \bar{\mathcal{L}}_i < \delta \right\}, D_{\text{noise}} = \left\{ (x_i, y_i) | \bar{\mathcal{L}}_i \geq \delta \right\}, \text{ where } \bar{\mathcal{L}}_i = \frac{\mathcal{L}_{CE}(x_i, y_i; \hat{\theta}_n)}{H(f_\theta(x_i))}. \quad (2)$$

Note that $H(f_\theta(x_i))$ represents the softmax entropy of sample $x_i$, and $\delta = \sum_{i \leq N} \bar{\mathcal{L}}_i / \sum_{i \leq N} H(f_\theta(x_i))$. In Figure 6, the black lines represent the threshold $\delta$, and the lower part of this line represents $D_{\text{clean}}$, whereas the upper part represents $D_{\text{noise}}$, respectively. Other discrimination results are depicted in Appendix E.

**De-noising module.** We design a de-noising module that is applicable in front of all prior de-biasing algorithms from the proposed criteria. As presented in Algorithm 3, this module comprises three simple steps. First, a *noise model* parameterized

---

**Algorithm 3** De-noising & De-biasing
1: Train network $f_{\theta_n}$ with CE loss $\mathcal{L}_{\text{CE}}$
2: Split $D_{\text{clean}}$ and $D_{\text{noise}}$ based on (2)
3: De-biasing using $D_{\text{clean}}$

---

by $\hat{\theta}_n$ is trained on the raw biased and noisy training sets. Then, we split $D_{\text{clean}}$ and $D_{\text{noise}}$ according to the criteria (2). After splitting, we run the de-biasing algorithm using $D_{\text{clean}}$.

## 6 EXPERIMENTS

We evaluated the proposed model for two types of benchmarks: synthetically generated bias (referred as *controlled biased benchmarks*) and raw bias (*real-world biased benchmarks*). We compared the results with the officially available recent de-biasing methods, RUBi (Cadene et al., 2019), Learned-MixinH (Clark et al., 2019), ReBias (Bahng et al., 2020), AFLite (Le Bras et al., 2020), LfF (Nam et al., 2020), REPAIR (Li & Vasconcelos, 2019) and the vanilla model. Each baseline was reproduced using the official codes for each algorithm (see Appendix C). Moreover, we optimized all algorithms based on the 10% of the biased training set. For the experiments, three types of convolutional neural networks were used with various regularization methods (batch normalization (Ioffe & Szegedy, 2015), dropout (Srivastava et al., 2014), and weight decaying (Moody & Hanson)). We report the detailed settings in Appendix A.1. In addition, we determined the test under flipped labels in MNIST variants with various noise ratios of 10%, 5%, and 0% to verify the side effects from noisy labels and check the protecting performance of the de-noising module (see Appendix A).

### 6.1 BENCHMARKS

#### 6.1.1 CONTROLLED BIASED BENCHMARKS

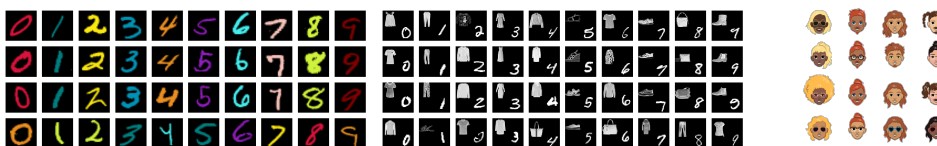

Figure 7: Examples of MNIST variants (Left: CM, Right: WM).      Figure 8: Cartoon.

To precisely examine the de-biasing performance, we generated controlled biased benchmarks using well-known data: MNIST (LeCun et al., 2010) and Cartoon (Royer et al., 2020). Various bias ratios, $|M|/|D| \in \{0.995, 0.99, 0.95\}$, were used for all the controlled tests. All benchmarks were divided into a biased training set $D_{tr}$, biased validation set $D_{val}$ (10% of $D_{tr}$), and test set $D_{te}$ comprising the majority $M$ and minority $m$ (see Appendix B.1 for detailed data construction rules.)

**MNIST Variants.** The MNIST dataset comprises a grayscale digit data with a handwritten style. We injected two different biases, color and different objects (specifically, Fashion MNIST (Xiao et al., 2017)), which are colored MNIST (CM) and watermarked MNIST (WM), respectively. This main task is classifying the *shape of the digit* without examining the biased attribute *color of the*

| Bias ratio | 99.5% | | | 99% | | | 95% | | |
|---|---|---|---|---|---|---|---|---|---|
| Test type | Val. $D_{val}$ | Major $M$ | Minor $m$ | Val. $D_{val}$ | Major $M$ | Minor $m$ | Val. $D_{val}$ | Major $M$ | Minor $m$ |
| **Colored MNIST** | | | | | | | | | |
| Vanilla | 99.49±0.01 | 99.97±0.05 | 26.58±1.36 | 99.37±0.05 | 99.97±0.05 | 51.97±0.84 | 99.46±0.07 | 99.87±0.05 | 90.92±0.40 |
| LearnedMixinH | 99.61±0.07 | 99.93±0.09 | 39.65±2.84 | 99.53±0.01 | 99.87±0.05 | 65.28±2.31 | 99.48±0.06 | 99.80±0.08 | 91.67±0.61 |
| RUBi | 99.48±0.01 | 99.93±0.09 | 29.36±1.17 | 99.43±0.02 | 99.90±0.08 | 56.97±1.87 | 99.56±0.01 | 99.90±0.00 | 92.09±1.11 |
| LfF | 98.35±0.17 | 98.05±0.17 | 58.36±3.25 | 95.75±2.25 | 95.56±2.72 | 81.42±3.98 | 99.15±0.05 | 99.21±0.08 | 96.55±0.26 |
| ReBias | 99.48±0.02 | 99.90±0.08 | 26.26±1.60 | 99.47±0.05 | 99.87±0.05 | 60.00±0.82 | 99.51±0.06 | 99.87±0.05 | 88.41±0.42 |
| AFLite | 98.39±0.77 | 98.78±0.68 | 29.59±1.80 | 98.97±0.16 | 99.27±0.26 | 57.04±1.41 | 99.04±0.02 | 99.37±0.19 | 89.77±0.56 |
| REPAIR | 99.65±0.05 | 99.93±0.09 | 42.14±2.75 | 99.52±0.03 | 99.83±0.05 | 70.31±2.42 | 99.51±0.01 | 99.87±0.05 | 91.39±1.08 |
| Ours | 99.13±0.07 | 99.07±0.17 | **96.50±0.33** | 98.91±0.05 | 99.24±0.19 | **97.76±0.36** | 98.93±0.19 | 99.37±0.33 | **97.42±0.28** |
| **Watermarked MNIST** | | | | | | | | | |
| Vanilla | 98.70±0.37 | 99.37±0.17 | 51.72±1.52 | 98.80±0.24 | 99.40±0.14 | 58.02±1.44 | 98.73±0.17 | 99.40±0.24 | 80.56±1.18 |
| LearnedMixinH | 99.27±0.05 | 99.80±0.00 | 65.95±0.32 | 99.17±0.25 | 99.74±0.05 | 77.32±0.92 | 99.50±0.16 | 99.77±0.05 | 85.35±1.24 |
| RUBi | 98.80±0.29 | 99.34±0.20 | 52.53±1.28 | 98.43±0.17 | 99.67±0.12 | 62.32±1.22 | 98.53±0.17 | 99.30±0.14 | 83.22±0.85 |
| LfF | 98.60±0.64 | 98.74±0.31 | 47.26±2.05 | 97.40±0.73 | 97.88±0.65 | 54.02±5.46 | 98.17±0.19 | 98.44±0.23 | 83.19±0.75 |
| ReBias | 98.93±0.17 | 99.44±0.05 | 48.82±4.72 | 98.53±0.47 | 99.34±0.09 | 54.98±0.61 | 98.83±0.05 | 99.54±0.20 | 80.62±1.89 |
| AFLite | 98.40±0.37 | 99.01±0.37 | 64.80±3.95 | 98.67±0.25 | 99.21±0.16 | 75.37±2.16 | 98.60±0.45 | 99.01±0.08 | 90.05±1.61 |
| REPAIR | 99.23±0.12 | 99.57±0.05 | 55.46±2.33 | 98.67±0.37 | 99.57±0.12 | 60.75±2.37 | 98.83±0.21 | 99.50±0.08 | 83.19±3.34 |
| Ours | 99.53±0.05 | 99.70±0.08 | **85.99±2.05** | 99.10±0.36 | 99.67±0.20 | **91.98±1.37** | 99.03±0.26 | 99.30±0.16 | **96.19±0.11** |
| **Cartoon** | | | | | | | | | |
| Vanilla | 99.80±0.04 | 99.99±0.02 | 46.44±1.92 | 99.42±0.11 | 99.99±0.02 | 56.25±4.64 | 99.14±0.23 | 99.95±0.05 | 78.26±2.93 |
| LearnedMixinH | 99.90±0.02 | 99.99±0.02 | 60.93±1.81 | 99.58±0.18 | 99.87±0.15 | 73.88±3.35 | 99.57±0.07 | 99.99±0.02 | 88.48±0.84 |
| RUBi | 99.72±0.07 | 99.82±0.16 | 43.88±1.65 | 98.97±0.57 | 99.71±0.33 | 52.61±0.99 | 98.27±0.63 | 99.57±0.48 | 70.11±3.28 |
| LfF | 99.72±0.04 | 99.99±0.02 | 45.42±3.08 | 99.44±0.07 | 99.91±0.10 | 55.17±2.90 | 99.14±0.18 | 99.81±0.18 | 81.02±3.01 |
| ReBias | 99.79±0.03 | 100.0±0.00 | 55.47±3.00 | 99.58±0.03 | 100.0±0.00 | 66.71±0.48 | 99.10±0.77 | 99.47±0.72 | 87.56±2.15 |
| AFLite | 99.38±0.06 | 99.36±0.46 | 51.49±1.52 | 98.96±0.47 | 99.54±0.34 | 55.77±0.53 | 98.28±0.17 | 99.70±0.17 | 72.17±1.99 |
| REPAIR | 99.87±0.37 | 100.0±0.00 | 47.59±2.41 | 99.47±0.02 | 100.0±0.00 | 55.41±1.94 | 99.16±0.24 | 99.94±0.09 | 80.34±3.47 |
| Ours | 99.01±0.46 | 98.62±0.63 | **80.22±5.17** | 99.06±0.55 | 98.96±0.42 | **83.28±6.14** | 98.54±0.62 | 98.29±0.60 | **94.23±1.35** |

Table 1: Average test accuracy and standard deviation (three runs) on experiments with the MNIST variants under various bias ratios. The best accuracy is indicated by bolded for each case.
*digit* or *Fashion MNIST objects*. As an example of the CM, class 0 is divided into *major* and *minor* sets colored in red and the others. Similarly, in WM, the *major* set of class 1 contains trouser, whereas the *minor* set contains other objects as watermarks at the top-left corner. Examples are depicted in Figure 7.

**Cartoon.** A Cartoon (CT) (Royer et al., 2020) is an avatar image with 18 corresponding attributes (e.g., hair style). We select two of them, hair color and face color, for the target and biased attributes. To create a bias, the *major* set is whose two attributes are bonded was subsampled. For example, in Figure 8, the face color of blond hair is black, except for a few.

### 6.1.2 REAL-WORLD BIASED BENCHMARKS

We used three well-known biased benchmarks to determine the de-biasing performance on complex benchmarks: Biased Action Recognition (BAR), CelebA, and ImageNet/ImageNet-A (see Appendix B.2 for details).

**Biased Action Recognition.** Biased action recognition (Bahng et al., 2020; Nam et al., 2020) is a motion classification set with a background bias. For example, in the class climbing, *major* and *Minor* samples have rockwall and ice cliff backgrounds, respectively.

**CelebA.** (Nam et al., 2020; Liu et al., 2015) CelebA has facial images with 40 binary attributes. Among these, we used heavy makeup and gender attributes for target and biased attributes. In the heavy makeup class, *major* is the female attribute, whereas male is the minority attribute.

**ImageNet/ImageNet-A.** ImageNet (Deng et al., 2009) is a well-known image classification dataset. However, (Hendrycks et al., 2019) claimed, a model trained on the ImageNet data easily fails to classify unusual backgrounds. This is because ImageNet has a biased background (e.g., almost all Frog images are considered in the swamp, which means *major* sample). As a de-biasing performance evaluation benchmark, we used ImageNet for training and to test ImageNet-A images with unusual backgrounds.

### 6.2 EVALUATION

**Controlled benchmarks without noisy labels.** The accuracy values of two *major* and *minor* sets are reported in Table 1. In particular, we first examined the accuracy of the vanilla algorithm, the ultimate baseline. Overall, the vanilla algorithm achieved nearly 100% accuracy for the *major* case, the vanilla failed to infer with an accuracy similar to that of the *major* case.

| Bias ratio / Noise ratio | 99% / 0% | | | | 99% / 5% | | | | 99% / 10% | | | |
|---|---|---|---|---|---|---|---|---|---|---|---|---|
| De-noising | Without de-noising | | With de-noising | | Without de-noising | | With de-noising | | Without de-noising | | With de-noising | |
| Test type | Major $M$ | Minor $m$ | Major $M$ | Minor $m$ | Major $M$ | Minor $m$ | Major $M$ | Minor $m$ | Major $M$ | Minor $m$ | Major $M$ | Minor $m$ |
| **Colored MNIST** | | | | | | | | | | | | |
| Vanilla | $99.97_{\pm 0.05}$ | $51.97_{\pm 0.84}$ | $98.94_{\pm 1.36}$ | $29.16_{\pm 0.11}$ | $99.90_{\pm 0.00}$ | $28.72_{\pm 0.95}$ | $99.97_{\pm 0.06}$ | $39.87_{\pm 1.69}$ | $100.0_{\pm 0.00}$ | $26.16_{\pm 2.67}$ | $100.0_{\pm 0.00}$ | $30.62_{\pm 2.57}$ |
| LfF | $95.56_{\pm 2.72}$ | $81.42_{\pm 3.98}$ | $94.65_{\pm 6.74}$ | $82.27_{\pm 11.2}$ | $7.96_{\pm 1.88}$ | $15.54_{\pm 5.43}$ | $99.77_{\pm 0.12}$ | $70.45_{\pm 2.78}$ | $0.00_{\pm 0.00}$ | $13.18_{\pm 3.29}$ | $99.68_{\pm 0.20}$ | $55.04_{\pm 7.16}$ |
| ReBias | $99.87_{\pm 0.05}$ | $60.00_{\pm 0.82}$ | $100.0_{\pm 0.00}$ | $47.28_{\pm 0.48}$ | $99.90_{\pm 0.00}$ | $30.57_{\pm 3.04}$ | $99.93_{\pm 0.06}$ | $40.21_{\pm 1.22}$ | $100.0_{\pm 0.00}$ | $20.85_{\pm 4.06}$ | $100.0_{\pm 0.00}$ | $31.41_{\pm 0.90}$ |
| REPAIR | $99.83_{\pm 0.05}$ | $70.31_{\pm 2.42}$ | $99.36_{\pm 0.07}$ | $51.75_{\pm 0.96}$ | $99.90_{\pm 0.00}$ | $22.48_{\pm 3.20}$ | $99.97_{\pm 0.06}$ | $45.64_{\pm 0.36}$ | $99.78_{\pm 0.37}$ | $11.94_{\pm 1.43}$ | $100.0_{\pm 0.00}$ | $34.43_{\pm 0.70}$ |
| Ours | $99.24_{\pm 0.19}$ | $\mathbf{97.76}_{\pm 0.36}$ | $99.76_{\pm 0.06}$ | $\mathbf{91.44}_{\pm 5.35}$ | $80.60_{\pm 0.00}$ | $\mathbf{87.50}_{\pm 0.74}$ | $99.47_{\pm 0.20}$ | $\mathbf{86.40}_{\pm 4.70}$ | $86.44_{\pm 3.16}$ | $\mathbf{82.13}_{\pm 4.87}$ | $99.78_{\pm 0.06}$ | $\mathbf{74.46}_{\pm 3.87}$ |
| **Watermarked MNIST** | | | | | | | | | | | | |
| Vanilla | $99.40_{\pm 0.14}$ | $58.02_{\pm 1.44}$ | $99.25_{\pm 0.23}$ | $47.17_{\pm 1.13}$ | $99.34_{\pm 0.06}$ | $56.10_{\pm 2.50}$ | $99.48_{\pm 0.04}$ | $52.98_{\pm 1.36}$ | $99.61_{\pm 0.10}$ | $53.42_{\pm 0.70}$ | $99.58_{\pm 0.05}$ | $53.59_{\pm 0.85}$ |
| LfF | $97.88_{\pm 0.65}$ | $54.02_{\pm 5.46}$ | $99.35_{\pm 0.33}$ | $44.84_{\pm 5.73}$ | $16.08_{\pm 1.83}$ | $15.17_{\pm 0.34}$ | $98.98_{\pm 0.30}$ | $53.56_{\pm 1.31}$ | $10.54_{\pm 0.64}$ | $10.42_{\pm 0.55}$ | $99.20_{\pm 0.14}$ | $53.97_{\pm 3.45}$ |
| ReBias | $99.34_{\pm 0.09}$ | $54.98_{\pm 0.61}$ | $99.30_{\pm 0.30}$ | $47.21_{\pm 1.39}$ | $99.28_{\pm 0.06}$ | $57.39_{\pm 0.78}$ | $99.28_{\pm 0.25}$ | $53.91_{\pm 1.03}$ | $99.48_{\pm 0.11}$ | $52.80_{\pm 2.21}$ | $99.55_{\pm 0.25}$ | $52.96_{\pm 3.01}$ |
| REPAIR | $99.57_{\pm 0.12}$ | $60.75_{\pm 2.37}$ | $99.45_{\pm 0.06}$ | $51.80_{\pm 0.32}$ | $99.48_{\pm 0.06}$ | $61.81_{\pm 1.89}$ | $99.64_{\pm 0.23}$ | $58.36_{\pm 3.46}$ | $99.52_{\pm 0.10}$ | $58.57_{\pm 3.13}$ | $99.64_{\pm 0.06}$ | $58.21_{\pm 1.03}$ |
| Ours | $99.67_{\pm 0.20}$ | $\mathbf{91.98}_{\pm 1.37}$ | $99.75_{\pm 0.07}$ | $\mathbf{64.66}_{\pm 7.09}$ | $52.37_{\pm 2.56}$ | $45.54_{\pm 3.18}$ | $99.77_{\pm 0.15}$ | $\mathbf{77.09}_{\pm 0.18}$ | $28.48_{\pm 7.47}$ | $26.11_{\pm 11.2}$ | $99.81_{\pm 0.10}$ | $\mathbf{79.05}_{\pm 3.84}$ |

Table 2: Average test accuracy and standard deviation (three runs) on experiments with the MNIST variants under a 1% bias ratio and $\{0\%, 5\%, 10\%\}$ noisy labels. The best accuracy is reported in **bolded** for the minority case.

De-biasing algorithms correctly classified the *major* case, similar to the vanilla case. However, for the *minor* case, these algorithms obtained higher accuracy values than that of the vanilla results (at most 31.78% ↑). Despite these de-biasing performances, they have a higher gap between *major* and *minor* cases (at least 36.69% under a 99.5% bias ratio). In contrast, the proposed method obtained the highest accuracy value for the *minor* case (at most 69.92% ↑) compared to the vanilla results. These results were obtained from the hyperparameter tuning with a biased validation set. In appendex D, we report the oracle results obtained with the optimal hyperparameters, which were tuned by using the majority-minority split validation set.

**Controlled benchmarks with noisy labels.** In Table 2, almost all algorithms failed to improve the minority accuracy, when labels are flipped, compared to the 0% noisy label case. For example, the proposed model's average performance declined from 98.50% to 84.05%, and the majority performance declined from 99.24% to 80.60% for the CM case with 5% noisy labels. However, these performance decrements were recovered from the de-noising module for all cases. For the 5% noise CM case, all cases obtained close to 100% accuracy for the majority case and improved the performance for the minority case. Among all methods, the proposed gradient-based method obtained 92.94% on average of the majority and minority test sets for the 5% noise of CM case, and improved by 7.83% in performance from the state-of-the-art algorithm, LfF at 85.11%. Moreover, we checked from the 0% noise case that the adverse effect of the de-noising module on de-biasing is $-3.9\%$ (99.50% → 95.60%) on CM case for the proposed method.

**Real-world benchmarks.** In Table 3, we report that real-world benchmarks can be biased by observing the lousy performance of the vanilla algorithm for all cases. The BAR case indicates that the proposed method gained the best performance on average among classes (a 5.06% increase from the vanilla algorithm and 2.53% increase from the second-best accuracy). As listed in other columns, the proposed method has helped improve vanilla by 3.64%

| Benchmarks | BAR | IN/IN-A | CelebA |
|---|---|---|---|
| Vanilla | $58.08_{\pm 0.01}$ | $34.55_{\pm 0.01}$ | $82.36_{\pm 1.02}$ |
| LfF | $57.35_{\pm 0.02}$ | $36.31_{\pm 0.01}$ | $81.13_{\pm 0.69}$ |
| REPAIR | $60.61_{\pm 0.00}$ | $35.46_{\pm 0.01}$ | $83.01_{\pm 0.62}$ |
| Ours | $\mathbf{63.14}_{\pm 0.01}$ | $\mathbf{37.42}_{\pm 0.01}$ | $\mathbf{86.01}_{\pm 0.49}$ |

Table 3: Test accuracy of real-world benchmarks averaged under three runs. The best one is reported in **bold font**.

and 2.87% for the CelebA and ImageNet/ImageNet-A cases, respectively. It also outperforms other methods by up to 1.11% and 3.00%, respectively. A detailed analysis is provided in Appendix F.

**Analysis.** We report the detailed various studies in Appendix H.

# 7 CONCLUSION

We presented a novel de-biased method by generating a mini-batch so that the trained model does not have prejudice in the inference. The proposed sampling metric is based on the gradient of each sample, which is mainly correlated with its familiarity. Through extensive experiments using various biased datasets, we demonstrated the effectiveness of the proposed method in several cases. Furthermore, we proposed a de-noising module that is easily applicable to all de-biasing algorithms and protects from the side effects of noisy labels. We hope this study proposed exploration to understand the feature bias problem, especially for resampling-based approaches and gradient-based data valuation areas of machine learning.

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

# A EXPERIMENT DETAILS

## A.1 BASIC SETTINGS

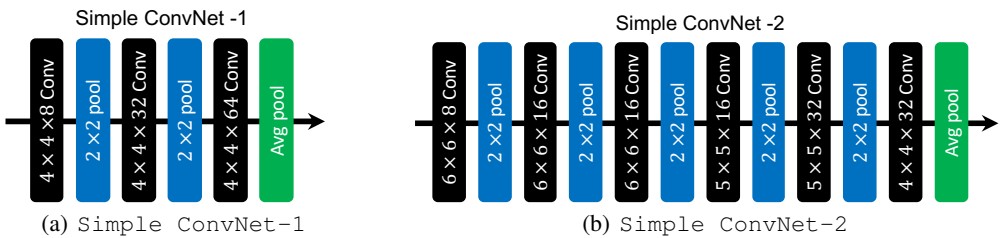

(a) `Simple ConvNet-1`    (b) `Simple ConvNet-2`

Figure 9: Simple convolutional networks for the variants of the MNIST task and Cartoon data.

**Network architecture.** We use three types of convolutional neural networks (CNNs) (see Figure 9 for two of them and the other is ResNet-18 (He et al., 2016)) for all benchmarks. For MNIST variants, three convolution layers with filter size 4 and average pooling with size 2 are used (see `Simple ConvNet-1`). For Cartoon, 5 convolution layers with filter size $[6, 6, 5, 5, 4]$ for each layer with average pooling size 2 are used (see `Simple ConvNet-2`). The real-world benchmarks are tested under pre-trained ResNet-18 provided by `Pytorch torchvision`, without freezing feature extractor. We apply batch normalization (Ioffe & Szegedy, 2015), dropout (Srivastava et al., 2014) for regularization, and utilize SGD optimizer with momentum and weight decay (Moody & Hanson).

| Parameters | Colored MNIST | Watermarked MNIST | Cartoon | Biased Action Recognition | CelebA | ImageNet/ImageNet-A |
|---|---|---|---|---|---|---|
| Network Type | Simple ConvNet-1 | Simple ConvNet-1 | Simple ConvNet-2 | Pretrained ResNet-18 | Pretrained ResNet-18 | Pretrained ResNet-18 |
| Learning rate | 0.01 | 0.01 | 0.002 | 0.001 | 0.001 | 0.001 |
| LR decay | 0.001 | 0.001 | 0.001 | 0.1 | 0.1 | 0.1 |
| LR decay epoch | [0.2,0.4,0.6,0.8] | [0.2,0.4,0.6,0.8] | [0.2,0.4,0.6,0.8] | [0.4,0.6,0.8] | [0.4,0.8] | [0.2,0.4,0.6,0.8] |
| Momentum | 0.8 | 0.8 | 0.9 | 0.3 | 0.3 | 0.3 |
| Weight decay | 0.001 | 0.001 | 0.001 | 5e-4 | 0.1 | 0.3 |
| Batch size | 256 | 256 | 256 | 64 | 64 | 64 |
| Epochs | 100 | 100 | 100 | 100 | 50 | 100 |
| GCE parameter $q$ | 0.7 | 0.7 | 0.7 | 0.7 | 0.7 | 0.7 |
| Dimension size $K$ | 100 | 100 | 100 | 500 | 500 | 500 |

Table 4: Hyperparameters for each task

**Hyperparameters.** The hyperparameters that we used are summarized in Table 4. We share basic hyperparameters (including `Network Type`, `Learning rate`, `Learning rate decay`, `Learning rate decay epoch`, `Momentum`, `Weight decay`, and `Batch size and Number of Epoch`) for all algorithms. Precisely, for the proposed method, we set GCE parameter $q$ and Dimension size $K$ for our method, as in Table 4.

**Validation.** We optimize hyperparameters using a validation set of $10\%$ of the training set and unseen at the training phase. Reported our results are obtained from the hyperparameters which shows a small loss for the validation set.

**Data augmentation.** Image data augmentation operations have been used frequently to improve generalization performance. We utilize three operations: `RandomResizedCrop` with scale $(0.9, 1.1)$, `ColorJitter` with $0.05$ hue and $0.05$ saturation, and `RandomRotation` with $(-10, 10)$ degree using `NEAREST` filling criterion. These augmentation operations are from `transforms` module in `torchvision`.

## A.2 TEST SETTINGS IN SECTION 3, 4, AND 5

**Ratio of minority samples analysis.** To analyze the correlation between $|m|/|D|$ and accuracy statistics, we generated a bias training set based on CM data generation in Appendix B.1. This tests are conducted on the experiment with difficulty $0.0001$.

**Gradient property test** To test our intuitions about gradients, we utilize part of CM data (includes class 0 and 1) with a bias ratio of $99.5\%$ and difficulty $0.0001$. Finally, we draw red arrows using the

lowest point and center point. This center point is the convergence point, and other points are random permuted loss values. All loss landscape figures are plotted by following (Li et al., 2018a).

# B  BENCHMARKS

## B.1  CONTROLLED BIASED BENCHMARKS

**MNIST Variants.** MNIST variants are modified data from gray-scaled hand-written digit MNIST images (LeCun et al., 2010). To inject biased attributes into the MNIST, we use two different attributes: color and object.

- **Colored MNIST** (CM) has been frequently used as a benchmark due to their simplicity (Kim et al., 2019; Bahng et al., 2020; Nam et al., 2020). The target of this benchmark is classifying the target attributes (shape of digits) not the biased attributes (color of digits). To make biased training set, all images are colored after up-scaling dimension from gray $\mathbb{R}^{28 \times 28}$ to 3D RGB $\mathbb{R}^{28 \times 28 \times 3}$. Uniformly sampled 3-dimensional 10 colors $C_c = \{R_c, G_c, B_c\}_{c=0}^9$ are allocated to each class $c$. The samples in class $c$ of set $M$ are colored by their allocated color $C_c$. The other samples are colored with the color vector $C_{c' \neq c}$ where $c' \neq c$ is uniformly sampled. Furthermore, to deviate correlation between the target and biased attributes, color for each sample $(x_i, y_i)$ is deviated using 3-dimensional Gaussian distribution $C_i = \mathcal{N}(C_{y_i}, \alpha I)$. This deviation is controlled by the difficulty parameter $\alpha \in \{0.0001, 0.0005\}$.
- **Watermarked MNIST** (WM) has a different type of bias. The target of this benchmark is the same with generic MNIST, classifying hand-written digits, while the unintended object (Fashion MNIST (Xiao et al., 2017) data) is located as a biased attributes. To locate Fashion MNIST, all images from MNIST are size-up scaled from $\mathbb{R}^{28 \times 28}$ to $\mathbb{R}^{56 \times 56}$. Set $M$ and $m$ includes samples $(x_i, y_i)$ where image $x_i$ is made with digit image $x_{\text{target}}$ and fashion object $x_{\text{bias}}$ whose label indices follow $y_i = y_{\text{target}} = y_{\text{bias}}$ and $y_i = y_{\text{target}} \neq y_{\text{bias}}$, respectively. To make deviation, we put images from $\delta$ pixels far from top and bottom of the canvas. $\delta$ is sampled from uniform distribution $\delta \sim \mathcal{U}(0, \alpha)$. We test two cases, $\alpha \in \{8, 16\}$.

**Cartoon.** Cartoon data composed of synthetically generated face image samples which have 18 attributes (e.g., `hair color`, `face color`) with multiple labels (e.g., `hair color` = $\{1, ..., 10\}$). Among 18 attributes, we choose `hair color` and `face color` attributes as the target attributes and the biased attributes. For simplicity, the size of each label is reduced from 9 to 4 and 10 to 4 by mapping 2 classes (similar colors) into 1 class, (e.g., original classes 1, 2 are mapped to the new class 1). We make set $M$ whose new target class and biased attributes following $c_{\text{target}} = c_{\text{bias}}$. Also, set $m$ are sampled among images following $c_{\text{target}} \neq c_{\text{bias}}$.

## B.2  REAL-WORLD BIASED BENCHMARKS

**Biased Action Recognition.** Biased action recognition data has $1,941$ motion images for training with 6-motions {`Climbing`, `Diving`, `Fishing`, `Racing`, `Throwing`, `Vaulting`} on various types of backgrounds (e.g., (`Climbing`, `Rockwall`)). The images in this benchmark include the frequent background of that action as a biased attributes for set $M$ and rare backgrounds for set $m$ (e.g., (`Climbing`, `Rockwall`) for $M$ and (`Climbing`, `Ice cliff`) for $m$). This benchmark is originated from (Bahng et al., 2020; Nam et al., 2020; Kim et al., 2021a).

**CelebA.** CelebA has $202,599$ face images with 40 binary attributes from $10,177$ identities. Among 40 attributes, we select two attributes (`HeavyMakeup`, `Male`) for target and biased attributes. The size of training subsets $M$ and $m$ are $130,410$, and $32,360$, respectively. Remains are used for evaluation. Splitting training and evaluation sets follow (Liu et al., 2015). This task reflects that a real-world benchmark is easy to be biased in some specific attributes, unintendedly. This benchmark is originated from (Nam et al., 2020).

**ImageNet/ImageNet-A.** ImageNet data has $1,000$ classes, and we utilize a subset of ImageNet, which has 9 super-classes for scalability (Bahng et al., 2020). Also, we evaluate using ImageNet-A (Hendrycks et al., 2019), which gathers the failure cases of ImageNet from web. As described in (Bahng et al., 2020; Hendrycks et al., 2019) dataset, "frequent background can be a biased

attributes." For example, frogs in the swamp have a higher proportion than other backgrounds (e.g., underwater). It indirectly represents that a de-biased model trained under ImageNet has better de-biasing performance when it has higher accuracy on ImageNet-A samples.

### B.3   NOISY LABELS

We inject $10\%$ noise labels into the basic configuration conducted in Section 5. Noise labels are uniform randomly flipped by following rules. At first, we randomly sampled a set of noise label candidates regardless of whether the source of the data samples are majority set or minority set. Flip the label $y_i \to \hat{y}_i$ uniform randomly, so that the label is not equal to the ground truth $\hat{y}_i \neq y_i$. Three noise label ratios, $\{90\%, 95\%, 1\%0\%\}$, were used. In order to train noised model $f_{\hat{\theta}_n}$, we utilize cross-entropy loss (not generalized cross-entropy loss). Also, to avoid over-fitting, we utilize three data augmentation operations: RandomResizedCrop, ColorJitter, and Random Rotation which are parameterized by $(0.9, 1.1), (0.05, 0.05)$ and $(-10, 10)$, respectively.

## C   BASELINES

**Vanilla**   As an ultimate baseline, which means without de-biasing module, we train a network called vanilla. As summarized before, we utilize frequently used regularization techniques and simple data augmentation operations (jitter, crop, and rotation).

**LearnedMixinH (Clark et al., 2019)**   LearnedMixinH was proposed by Clark et al for VQA task. This method of de-biasing is composed of biased (on the question) and unbiased (on both the image and the question) networks. We generate a biased network in our experiment by simply training on the biased data set. This is based on the argument in LfF (Nam et al., 2020), which states that biased models learn biased information first, because it is easier to learn than the target attribute. The biased model outputs two values: its softmax output and a bias prediction value for each given sample. Ultimately, the de-biased model trained under the cross-entropy loss with regularizer of two above values from the biased model. We implement this algorithm by inheriting the authors' official code and unofficial code made by the authors (Bahng et al., 2020).

**RUBi (Cadene et al., 2019)**   Similar with LearnedMixinH, RUBi also targets to solve VQA tasks. We also utilize the argument from LfF (Nam et al., 2020), to train the biased network. Instead using two values of the biased model, the authors of this paper utilized sigmoid outputs of the biased model and give weights to the de-biased model's output, when calculating the cross-entropy loss. This weighting mechanism gives weighted punishements by followings. The biased model outputs highly and lower biased sigmoid output when `major` and `minor` samples are given. By multiplying sigmoid values to the softmax output of the de-biased model, the de-biased model suffers higher loss from the `minor` samples. We implement this method using the authors official code and unofficial code from (Bahng et al., 2020).

**ReBias (Bahng et al., 2020)**   ReBias is a bi-level optimization-based de-biasing algorithm proposed by Bahng et al. ReBias trains the biased and de-biased model simultaneously. The de-biased model updated its cross-entropy loss and additional regularizer using Hilbert-Schmidt Independence Criterion (HSIC). Here, the biased model is composed of large convolutional filters to capture the bias information better, but we ignore this for a fair comparison with same network. We utilize official code from the authors.

**LfF (Nam et al., 2020)**   The main module of this algorithm is two-fold: (i) The biased model trained on Generalized cross-entropy (GCE), which highlights the `major` samples. (ii) The de-biased model trained on the weighted loss where weight is computed by the cross-entropy outputs from each. From GCE, the cross entropy loss loss of the `minor` samples increases. We utilize official code from authors.

**AFLite (Le Bras et al., 2020)**   AFLite is another type of resampling approach. AFLite trains the biased model multiple times and selects samples whose accuracy is low. At the de-biasing phase, the de-biased model is trainend on the sub-sampled new training set. Because authors offer the official code for a synthetic data, we manually reproduce their idea.

**REPAIR (Li & Vasconcelos, 2019)** REPAIR is a re-sampling-based de-biasing algorithm proposed by Li et al. REPAIR reconfigure the training set based on the weight parameter $w$ obtained by the adversarial training. As a first step, the biased model and the adversary are trained simultaneously. The adversary computes per sample weight $w$ so that the biased model gets a larger value for the minorities. REPAIR utilized $w$ to reconstruct the repaired training set. The authors proposed four types of re-configuring methods with given parameter $k$: REPAIR-T (Thresholding, sampling above $k$), REPAIR-R (Ranking, Sampling top-$k\%$), REPAIR-PR (Per-class Ranking), REPAIR-S (Sampling, reject with probability $1 - w$). We utilize REPAIR-S among these variants to fairly compare our sampling mechanism (without hyperparameter $k$ case). We implement REPAIR using the official code from the authors. Additionally, note that we obtained $w$ by training the classifier at the adversarial training phase without using biased information, such as RGB values in the case of CM, where as the official code used the biased information. This is enabled by the LfF (Nam et al., 2020) argument, which state that the network is biased toward a biased attribute that is easier to learn than the target.

## D RESULT: CONTROLLED BIASED BENCHMARKS (ORACLE VALIDATION)

These results are obtained from the tuned hyperparameters, which are optimized by using the labels related to the biased attributes $b$. As we can see, the proposed method obtains the best performance in terms of the average accuracy of majority and minority cases. Moreover, other works also successfully improves their results. However, it is difficult to obtain a validation set which is discriminated between majority and minority samples. Therefore, we can conclude that the proposed method obtains when we can access to the discriminated dataset between majority and minority cases, which is difficult in practice.

| Bias ratio | 99.5% | | 99% | | 98% | | 95% | |
|---|---|---|---|---|---|---|---|---|
| Test type | Major | Minor | Major | Minor | Major | Minor | Major | Minor |
| **Difficulty : 0.0001** | | | | | | | | |
| Vanilla | $99.97_{\pm0.05}$ | $29.22_{\pm1.87}$ | $99.93_{\pm0.09}$ | $56.09_{\pm2.19}$ | $99.87_{\pm0.09}$ | $72.92_{\pm2.54}$ | $99.90_{\pm0.00}$ | $89.60_{\pm0.51}$ |
| LearnedMixinH (Clark et al., 2019) | $99.93_{\pm0.09}$ | $36.27_{\pm2.79}$ | $100.0_{\pm0.00}$ | $60.20_{\pm4.29}$ | $96.77_{\pm4.50}$ | $74.63_{\pm1.08}$ | $96.44_{\pm4.96}$ | $88.64_{\pm3.95}$ |
| RUBi (Cadene et al., 2019) | $99.97_{\pm0.05}$ | $31.66_{\pm1.12}$ | $99.90_{\pm0.08}$ | $57.23_{\pm2.49}$ | $99.90_{\pm0.08}$ | $77.77_{\pm0.80}$ | $99.87_{\pm0.05}$ | $89.99_{\pm0.44}$ |
| LfF (Nam et al., 2020) | $99.63_{\pm0.48}$ | $56.00_{\pm3.72}$ | $97.43_{\pm0.12}$ | $84.05_{\pm2.14}$ | $95.05_{\pm0.44}$ | $92.23_{\pm1.04}$ | $98.03_{\pm0.25}$ | $95.29_{\pm0.49}$ |
| REPAIR (Li & Vasconcelos, 2019) | $99.77_{\pm0.17}$ | $64.97_{\pm0.30}$ | $99.90_{\pm0.08}$ | $79.12_{\pm0.35}$ | $99.71_{\pm0.08}$ | $87.74_{\pm0.63}$ | $99.77_{\pm0.05}$ | $92.96_{\pm0.21}$ |
| Ours | $\mathbf{99.06_{\pm0.24}}$ | $\mathbf{97.22_{\pm0.19}}$ | $\mathbf{99.19_{\pm0.12}}$ | $\mathbf{98.19_{\pm0.16}}$ | $\mathbf{98.74_{\pm0.14}}$ | $\mathbf{98.57_{\pm0.19}}$ | $\mathbf{99.16_{\pm0.18}}$ | $\mathbf{98.58_{\pm0.05}}$ |
| **Difficulty : 0.0005** | | | | | | | | |
| Vanilla | $99.97_{\pm0.05}$ | $28.90_{\pm1.39}$ | $99.93_{\pm0.09}$ | $55.85_{\pm2.30}$ | $99.87_{\pm0.09}$ | $73.16_{\pm2.42}$ | $99.90_{\pm0.00}$ | $89.52_{\pm0.71}$ |
| LearnedMixinH (Clark et al., 2019) | $99.93_{\pm0.09}$ | $36.42_{\pm2.70}$ | $100.0_{\pm0.00}$ | $60.44_{\pm4.30}$ | $96.77_{\pm4.50}$ | $74.47_{\pm1.24}$ | $96.48_{\pm4.98}$ | $88.71_{\pm3.99}$ |
| RUBi (Cadene et al., 2019) | $99.97_{\pm0.05}$ | $31.90_{\pm1.27}$ | $99.90_{\pm0.08}$ | $57.21_{\pm2.48}$ | $99.90_{\pm0.08}$ | $77.78_{\pm0.95}$ | $99.87_{\pm0.05}$ | $89.90_{\pm0.44}$ |
| LfF (Nam et al., 2020) | $98.63_{\pm0.48}$ | $56.00_{\pm3.70}$ | $97.46_{\pm0.14}$ | $84.03_{\pm2.12}$ | $95.02_{\pm0.45}$ | $92.24_{\pm1.00}$ | $98.00_{\pm0.24}$ | $95.29_{\pm0.51}$ |
| REPAIR (Li & Vasconcelos, 2019) | $99.77_{\pm0.17}$ | $65.10_{\pm0.31}$ | $99.93_{\pm0.09}$ | $79.06_{\pm0.44}$ | $99.68_{\pm0.12}$ | $87.81_{\pm0.56}$ | $99.84_{\pm0.05}$ | $92.96_{\pm0.16}$ |
| Ours | $\mathbf{99.15_{\pm0.18}}$ | $\mathbf{97.23_{\pm0.21}}$ | $\mathbf{98.99_{\pm0.05}}$ | $\mathbf{98.21_{\pm0.15}}$ | $\mathbf{98.64_{\pm0.27}}$ | $\mathbf{98.58_{\pm0.11}}$ | $\mathbf{99.06_{\pm0.12}}$ | $\mathbf{98.56_{\pm0.04}}$ |

Table 5: Average test accuracy and standard deviation (3 independent runs) on experiments with Colored MNIST data under various bias ratio. The best accuracy on average of `Major` and `Minor` test is reported in **bold font**.

| Bias ratio | 99.5% | | 99% | | 98% | | 95% | |
|---|---|---|---|---|---|---|---|---|
| Test type | Major | Minor | Major | Minor | Major | Minor | Major | Minor |
| **Difficulty : 8** | | | | | | | | |
| Vanilla | $98.99_{\pm0.20}$ | $47.52_{\pm1.45}$ | $99.15_{\pm0.30}$ | $53.74_{\pm0.59}$ | $99.19_{\pm0.25}$ | $66.77_{\pm0.44}$ | $99.55_{\pm0.09}$ | $79.34_{\pm2.90}$ |
| LearnedMixinH (Clark et al., 2019) | $99.48_{\pm0.09}$ | $68.49_{\pm1.73}$ | $99.41_{\pm0.14}$ | $74.50_{\pm1.07}$ | $99.71_{\pm0.08}$ | $84.02_{\pm1.70}$ | $99.81_{\pm0.14}$ | $91.90_{\pm0.49}$ |
| RUBi (Cadene et al., 2019) | $99.48_{\pm0.14}$ | $52.79_{\pm2.55}$ | $99.06_{\pm0.30}$ | $56.91_{\pm1.18}$ | $99.10_{\pm0.25}$ | $64.31_{\pm2.16}$ | $99.48_{\pm0.20}$ | $81.56_{\pm1.32}$ |
| LfF (Nam et al., 2020) | $99.22_{\pm0.58}$ | $49.03_{\pm2.11}$ | $98.47_{\pm0.05}$ | $56.13_{\pm4.00}$ | $98.61_{\pm0.23}$ | $67.21_{\pm2.19}$ | $98.65_{\pm0.36}$ | $86.43_{\pm1.27}$ |
| REPAIR (Li & Vasconcelos, 2019) | $98.96_{\pm0.09}$ | $70.75_{\pm2.01}$ | $98.93_{\pm0.08}$ | $77.02_{\pm1.19}$ | $99.39_{\pm0.12}$ | $83.94_{\pm1.29}$ | $99.68_{\pm0.09}$ | $91.30_{\pm0.28}$ |
| Ours | $\mathbf{99.35_{\pm0.23}}$ | $\mathbf{83.10_{\pm1.35}}$ | $\mathbf{99.45_{\pm0.20}}$ | $\mathbf{91.09_{\pm1.96}}$ | $\mathbf{98.87_{\pm0.12}}$ | $\mathbf{93.70_{\pm0.53}}$ | $\mathbf{99.39_{\pm0.25}}$ | $\mathbf{96.16_{\pm0.33}}$ |
| **Difficulty : 16** | | | | | | | | |
| Vanilla | $99.09_{\pm0.33}$ | $48.18_{\pm1.08}$ | $99.09_{\pm0.12}$ | $54.15_{\pm1.02}$ | $99.16_{\pm0.25}$ | $65.61_{\pm0.56}$ | $99.61_{\pm0.08}$ | $79.16_{\pm1.95}$ |
| LearnedMixinH (Clark et al., 2019) | $99.41_{\pm0.16}$ | $69.80_{\pm1.39}$ | $99.84_{\pm0.05}$ | $73.13_{\pm1.11}$ | $99.77_{\pm0.16}$ | $85.17_{\pm1.00}$ | $99.65_{\pm0.16}$ | $92.25_{\pm0.59}$ |
| RUBi (Cadene et al., 2019) | $99.15_{\pm0.12}$ | $51.89_{\pm0.63}$ | $98.99_{\pm0.18}$ | $54.94_{\pm1.16}$ | $99.35_{\pm0.12}$ | $65.99_{\pm0.94}$ | $99.10_{\pm0.28}$ | $81.68_{\pm0.19}$ |
| LfF (Nam et al., 2020) | $98.37_{\pm0.46}$ | $50.40_{\pm3.20}$ | $98.47_{\pm0.39}$ | $56.97_{\pm1.89}$ | $98.45_{\pm0.36}$ | $66.56_{\pm2.79}$ | $97.84_{\pm0.32}$ | $85.18_{\pm0.92}$ |
| REPAIR (Li & Vasconcelos, 2019) | $99.02_{\pm0.08}$ | $70.76_{\pm1.83}$ | $99.02_{\pm0.24}$ | $75.89_{\pm1.37}$ | $99.10_{\pm0.41}$ | $84.29_{\pm0.74}$ | $99.52_{\pm0.24}$ | $88.71_{\pm2.50}$ |
| Ours | $\mathbf{99.54_{\pm0.12}}$ | $\mathbf{84.07_{\pm1.79}}$ | $\mathbf{98.86_{\pm0.30}}$ | $\mathbf{90.46_{\pm0.81}}$ | $\mathbf{99.26_{\pm0.16}}$ | $\mathbf{94.25_{\pm1.04}}$ | $\mathbf{99.16_{\pm0.40}}$ | $\mathbf{96.14_{\pm0.37}}$ |

Table 6: Average test accuracy and standard deviation (3 independent runs) on experiments with Watermarked MNIST data under various bias ratio. The best accuracy on average of `Major` and `Minor` test is reported in **bold font**.

| Bias ratio | 99.5% | | 99% | | 98% | | 95% | |
|---|---|---|---|---|---|---|---|---|
| Test type | Major | Minor | Major | Minor | Major | Minor | Major | Minor |
| Vanilla | $100.0_{\pm 0.00}$ | $33.82_{\pm 2.70}$ | $99.97_{\pm 0.02}$ | $54.55_{\pm 5.04}$ | $99.76_{\pm 0.26}$ | $72.45_{\pm 2.72}$ | $99.84_{\pm 0.21}$ | $90.31_{\pm 1.35}$ |
| LearnedMixinH (Clark et al., 2019) | $100.0_{\pm 0.00}$ | $33.82_{\pm 2.70}$ | $99.96_{\pm 0.00}$ | $48.80_{\pm 0.00}$ | $99.92_{\pm 0.00}$ | $70.27_{\pm 0.00}$ | $99.58_{\pm 0.00}$ | $88.14_{\pm 0.00}$ |
| RUBi (Cadene et al., 2019) | $100.0_{\pm 0.00}$ | $22.51_{\pm 7.67}$ | $99.99_{\pm 0.02}$ | $46.76_{\pm 7.24}$ | $99.87_{\pm 0.13}$ | $68.54_{\pm 2.14}$ | $99.91_{\pm 0.06}$ | $87.01_{\pm 0.55}$ |
| LfF (Nam et al., 2020) | $90.56_{\pm 7.89}$ | $56.85_{\pm 2.43}$ | $81.77_{\pm 5.17}$ | $76.97_{\pm 4.34}$ | $78.84_{\pm 7.39}$ | $77.23_{\pm 2.92}$ | $85.93_{\pm 5.49}$ | $80.02_{\pm 3.68}$ |
| REPAIR (Li & Vasconcelos, 2019) | $99.95_{\pm 0.02}$ | $60.36_{\pm 7.97}$ | $99.95_{\pm 0.02}$ | $80.06_{\pm 4.51}$ | $99.96_{\pm 0.03}$ | $88.74_{\pm 2.58}$ | $99.90_{\pm 0.05}$ | $94.08_{\pm 1.31}$ |
| Ours | $\mathbf{97.94}_{\pm 1.31}$ | $\mathbf{92.53}_{\pm 2.66}$ | $\mathbf{98.33}_{\pm 0.16}$ | $\mathbf{93.75}_{\pm 1.25}$ | $\mathbf{96.95}_{\pm 1.78}$ | $\mathbf{92.57}_{\pm 2.69}$ | $\mathbf{98.61}_{\pm 0.79}$ | $\mathbf{95.24}_{\pm 0.93}$ |

Table 7: Average test accuracy and standard deviation (3 independent runs) on experiments with Cartoon data under various bias ratio. The best accuracy on average of Major and Minor test is reported in **bold font**.

# E    SPLITTING PERFORMANCE OF DE-NOISING MODULE

Discriminativeness power of the de-noising module between majority and minority cases on various noisy label ratio and bias ratio $|m|/|D|$. As we can see for all cases, our de-noising module with compensated loss criterion can distinguish the minority samples and noisy labels. In addition, when there is no noisy cases, ($0\%$ cases) the proposed de-noising module lose some valuable clean minority samples. All cases are run with difficulty parameter $0.0001$ and $8$ for CM and WM cases, respectively.

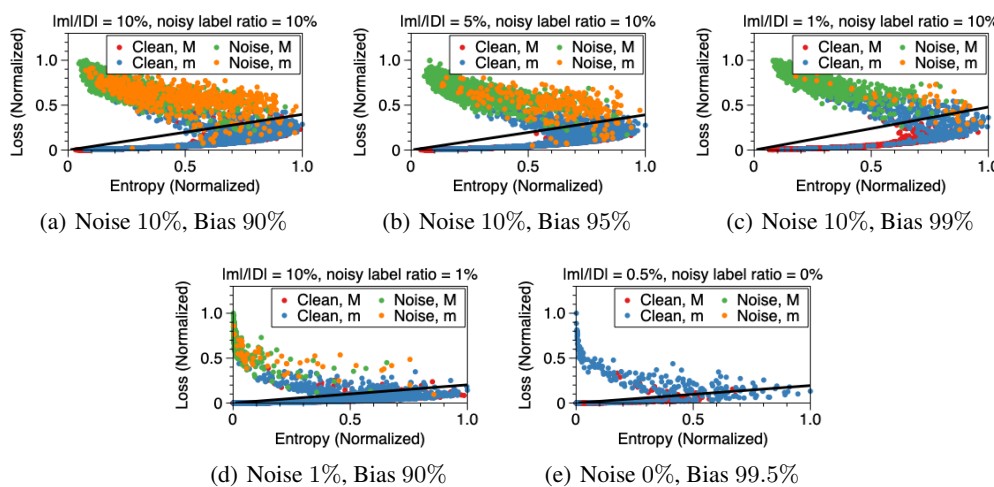

Figure 10: De-noising module and threshold plots for various conditions. Data= CM, Noise=$\{0\%, 1\%, 10\%\}$ and Bias=$\{99.5\%, 1\%, 5\%, 10\%\}$.

# F    REAL-WORLD BENCHMARK: DETAIL ACCURACY

As in all cases, the proposed method obtains the best performance on average of all cases, e.g., CelebA on average of majority and minority cases. Also, our BAR results fails to obtain state-of-the-art performance in some classes, but get the best on average of all classes.

| Benchmark | CelebA | | |
|---|---|---|---|
| Type | Major | Minor | Avg. |
| Vanilla | $93.16_{\pm 0.75}$ | $71.56_{\pm 2.41}$ | $82.36_{\pm 1.02}$ |
| LfF | $92.60_{\pm 0.58}$ | $69.67_{\pm 1.88}$ | $81.13_{\pm 0.69}$ |
| REPAIR | $92.54_{\pm 0.52}$ | $73.48_{\pm 1.75}$ | $83.01_{\pm 0.62}$ |
| Ours | $\mathbf{88.18}_{\pm 0.83}$ | $\mathbf{83.83}_{\pm 1.74}$ | $\mathbf{86.01}_{\pm 0.49}$ |

Table 8: Test accuracy of real-world benchmarks averaged under three runs. The best accuracy is reported in **bold font**.

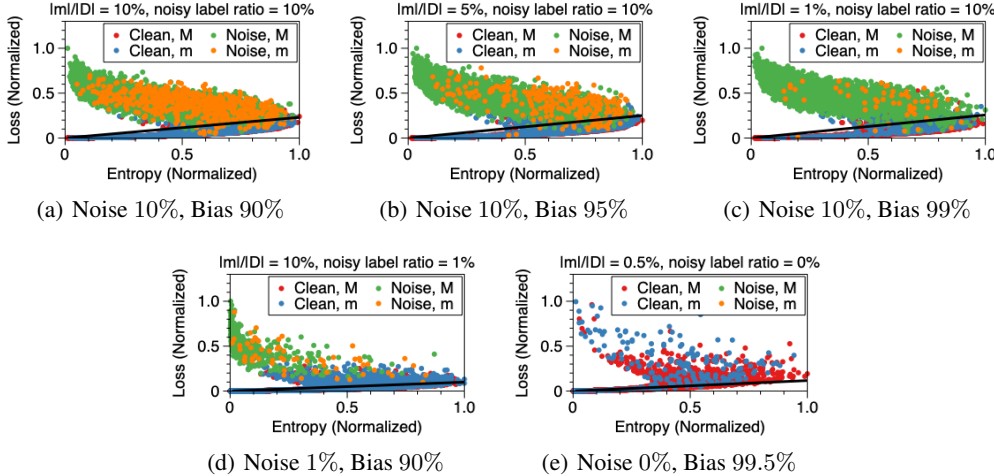

(a) Noise 10%, Bias 90%    (b) Noise 10%, Bias 95%    (c) Noise 10%, Bias 99%

(d) Noise 1%, Bias 90%    (e) Noise 0%, Bias 99.5%

Figure 11: De-noising module and threshold plots for various conditions. Data= WM, Noise={0%, 1%, 10%} and Bias={99.5%, 1%, 5%, 10%}.

| Benchmark | IN/IN-A |
|---|---|
| Type | Avg. |
| Vanilla | $34.55_{\pm 0.01}$ |
| LfF | $36.31_{\pm 0.01}$ |
| REPAIR | $35.46_{\pm 0.01}$ |
| Ours | $\mathbf{37.42}_{\pm 0.01}$ |

Table 9: Test accuracy of real-world benchmarks averaged under three runs. The best accuracy is reported in **bold font**.

| Benchmark | Biased action recognition | | | | | | |
|---|---|---|---|---|---|---|---|
| Type | Climbing | Diving | Fishing | Racing | Throwing | Vaulting | Avg. |
| Vanilla | $67.62_{\pm 4.67}$ | $29.98_{\pm 3.14}$ | $66.67_{\pm 1.94}$ | $80.30_{\pm 2.70}$ | $35.29_{\pm 1.66}$ | $69.72_{\pm 5.00}$ | $58.08_{\pm 0.01}$ |
| LfF | $68.57_{\pm 3.56}$ | $32.91_{\pm 5.19}$ | $61.11_{\pm 2.97}$ | $78.28_{\pm 2.92}$ | $32.94_{\pm 2.88}$ | $70.48_{\pm 1.57}$ | $57.35_{\pm 0.02}$ |
| REPAIR | $66.98_{\pm 1.62}$ | $38.36_{\pm 5.41}$ | $70.63_{\pm 1.12}$ | $\mathbf{80.81}_{\pm 0.94}$ | $37.65_{\pm 0.96}$ | $\mathbf{71.25}_{\pm 3.43}$ | $60.61_{\pm 0.00}$ |
| Ours | $\mathbf{71.43}_{\pm 0.78}$ | $\mathbf{45.91}_{\pm 0.51}$ | $\mathbf{72.22}_{\pm 2.97}$ | $79.80_{\pm 2.17}$ | $\mathbf{42.35}_{\pm 0.00}$ | $68.70_{\pm 3.30}$ | $\mathbf{63.14}_{\pm 0.01}$ |

Table 10: Test accuracy of real-world benchmarks averaged under three runs. The best accuracy is reported in **bold font** in each column.

## G    VON MISES-FISHER DISTRIBUTION FOR GRADIENTS

The von Mises-Fisher (vMF) distribution is a probability distribution on the hypersphere $S^{d-1} \subset \mathbb{R}^d$. The probability density function of the vMF distribution for the unit vector $v$ is given by:

$$f_d(v; u, \kappa) = C_d(\kappa)e^{(\kappa u^\top v)},$$

where $\kappa$ and $u$ are called *concentration parameter* and *mean direction*, respectively. The normalization constant $C_d(\kappa)$ is equal to $C_d(\kappa) = \frac{\kappa^{p/2-1}}{(2\pi)^{p/2} I_{p/2-1}(\kappa)}$, where $I_t$ denotes the modified Bessel function of the first kind at order $t$.

As in (Banerjee et al., 2005), when $N$ samples from the vMF distribution are given as $v_i$, the maximum likelihood estimates solutions $\hat{u}$ and $\hat{\kappa}$ can be obtained from $\hat{u} = \frac{\sum_{i=1}^N v_i}{\|\sum_{i=1}^N v_i\|}$ and $\hat{\kappa} \approx \frac{\bar{r}(d-\bar{r}^2)}{1-\bar{r}^2}$, where $\bar{r} = \frac{\|\sum_{i=1}^N v_i\|}{N}$.

To interpret gradient vectors as vMF distribution, given vectors have to be unit vectors: $\overline{\nabla \theta_i} = \frac{\nabla \theta_i}{\|\nabla \theta_i\|}$. Therefore, the approximated vMF distribution of the gradients can be obtained as follows:

$$f_d(\overline{\nabla \theta_i}; \hat{u}, \hat{\kappa}) \text{ where } \hat{u} = \frac{\sum_{i=1}^{N} \overline{\nabla \theta_i}}{\| \sum_{i=1}^{N} \overline{\nabla \theta_i} \|}, \quad \hat{\kappa} \approx \frac{\bar{r}(d - \bar{r}^2)}{(1 - \bar{r}^2)}, \text{ and } \bar{r} = \frac{\| \sum_{i=1}^{N} \overline{\nabla \theta_i} \|}{N}.$$

## H   ANALYSIS

We conducted a few studies to analyze our method: hyperparameter sensitivity, (target,bias)-pairwise accuracy, pairwise sampling probability, ablation study on $\lambda$, adjusting objective results based on our metric, class activation map, unbiased case result, training time, results on naive metrics, and impact of GCE. Controlled benchmarks were used for analysis.

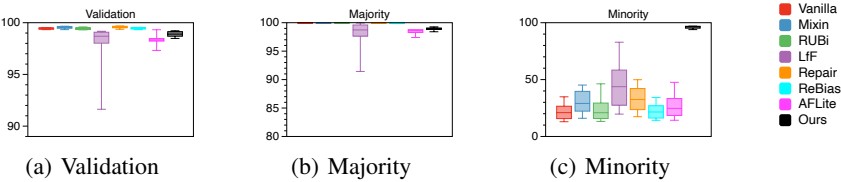

Figure 12: Hyperparameter sensitivity. We train models on CM with diffculty 0.0001 under learning rate $\eta \in \{0.005, 0.01, 0.02\}$ and epoch $e \in \{50, 100\}$.

**Hyperparameter sensitivity.** In Figure 12, accuracy results of the models trained on the various learning rate $\eta \in \{0.005, 0.01, 0.02\}$ and epochs $e \in \{50, 100\}$ were depicted. It shows that the prediction performance of all algorithms has higher accuracies on biased validation sets and majority set for various learning rates and epochs. However, in minority case 12(c), ours have a less sensitive result on minority cases by seeing error bar's length.

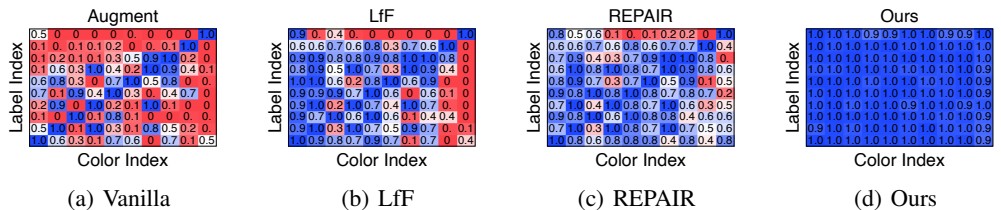

Figure 13: Accuracy on each (color, label)-pair. Red color corresponds to bad accuracy, while blue color denotes higher accuracy.

**(Target, Bias)-pairwise accuracy.** In Figure 13, pairwise accuracy matrices were plotted using various de-biasing methods. This represents the prediction result of 10 *Major* (diagonal) and 90 *Minor* pairs. First, vanilla fails to infer *Minor* pairs, but it infers *Major* accurately. Similar to vanilla, other algorithms, including ours, achieve almost accurate results for the *Major* pairs. On the other hand, based on the perspective of *minority*, other algorithms fail to infer. Among the de-biasing methods, our method has been approximately 100% accurate for almost all the pairs.

**Pairwise sampling probability.** To understand the previous pairwise result, we checked the sampling probability of our method. In Figure 14, each subfigure indicates pairwise sampling probability for all the metrics: $p_M(i) = \frac{1/M(i)}{\sum_i (1/M(i))}, p_D(i) = \frac{1/D(i)}{\sum_i (1/D(i))}$, and $p_s(i) = 1/S(i)$. In Figure 14(a), the raw case is highly bonded between the shape and color (see diagonal). In $p_M(i)$ and $p_D(i)$ cases, both are well mixed compared to without case. In Figure 14(d), $p_s(i)$ has the most uniformized sampling probability. Red and yellow indicators denote complementary effect of both scores.

**Ablation study on $\lambda$.** In the definition of the score $S(i)$, we set the hyperparameter $\lambda$ to make balance between source scores $M(i)$ and $D(i)$. In order to confirm the effect of $\lambda$, the experimental results

| $\lambda$ | 0 | 1 |
|---|---|---|
| Acc. | 97.46 / 97.41 | 97.97 / 97.10 |
| $\lambda$ | 10 | 100 |
| Acc. | 97.97 / 96.99 | 97.16 / 95.64 |

Table 11: Ablation study on $\lambda$

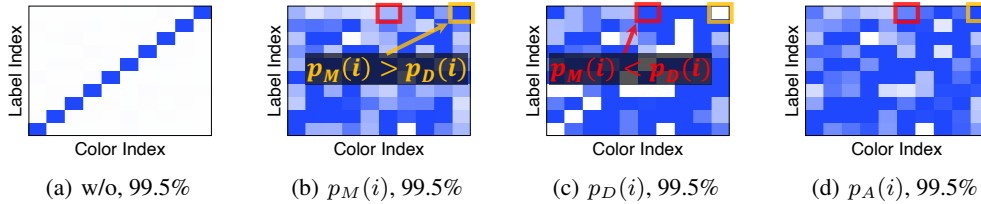

| (a) w/o, 99.5% | (b) $p_M(i)$, 99.5% | (c) $p_D(i)$, 99.5% | (d) $p_A(i)$, 99.5% |

Figure 14: Pairwise sampling probability under colored MNIST with 0.0001 difficulty. Blue color indicates higher sampling probability.

are reported from various values of $\lambda = \{0, 1, 10, 100\}$ on CM with 99% bias without noise. As in Table 11, there is no remarkable performance difference depending on the $\lambda$, so it could be observed that it is robust to the hyperparameter $\lambda$.

**Adjusting objective vs Resampling.** As an ablation study, we clarified the power of our gradient-based metric by reporting the results of adjusting objective leveraged by our $p_s(i)$. We conducted tests under two controlled biased benchmarks, colored MNIST and watermarked MNIST with difficulty 0.0001 and 8. $p_s(i)$, normalized by the mean value, $\frac{p_s(i)}{\sum_i^N p_s(i)/N}$, is used for the weights. Our gradient-based score

| Benchmark | C. M. | W. M. |
|---|---|---|
| Type | Avg. | Avg. |
| Vanilla | 64.56 | 73.26 |
| LfF | 77.82 | 74.13 |
| REPAIR | 82.37 | 84.86 |
| Ours† | 95.33 | 86.67 |
| Ours | 98.14 | 91.23 |

outperforms others in the adjusting objective (dagger mark †) and resampling cases. Compared with our gradient-based ways, the adjusting objective methods achieve lower performance. This implies that, as argued in (An et al., 2020), the adjusting objective-based approach suffers from training sensitivity.

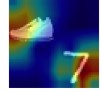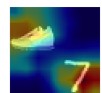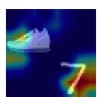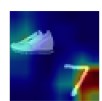 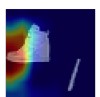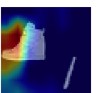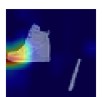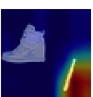

(a) Vanilla, $M$ (b) LfF, $M$ (c) Repair, $M$(d) Ours, $M$    (e) Vanilla, $m$ (f) LfF, $m$ (g) Repair, $m$(h) Ours, $m$

Figure 15: Plots of class activation map (CAM) for each method of same inputs, sampled from the *Major (M)* and *Minor (m)* sets.

**What de-biased model learn.** The main goal of de-biasing algorithms is to make the de-biased model not learn the biased feature. To check whether the biased models satisfy this goal, we evaluate two tests: (1) random color test for CM, and (2) CAM for WM.

We construct test dataset by uniformly randomly sample colors for all samples RGB $\sim \mathcal{U}(0, 1)^3$. As in Table 12, the result of ours show that it outputs based on the shape whether the training set is biased or not.

| Algorithm | Accuracy |
|---|---|
| Vanilla | 50.80 |
| LfF | 85.04 |
| ReBias | 51.49 |
| REPAIR | 53.28 |
| Ours | 95.44 |

Table 12: Random color test results on CM

Additionally, we conduct CAM test for WM dataset. CAM (Selvaraju et al., 2017) has been used as a tool to explain "where a model is focusing on?" In Figure 15, we state the CAM results for watermarked MNIST samples obtained from the *Major* (**Left**) and *Minor* (**Right**) sets. In the top row (for *Major*), our model better ignores the biased object (fashion object) than the other methods. Even for *Minor*, only our model determines the target `digit`, while the rest focuses on the biased object (Ankle boots).

**Unbiased / Rarely biased case.** To analyze the general side effect of additional modules, performance degradation, we check the accuracy of unbiased (especially, $10\% = 1/(\text{# of classes})$ biased case) and rarely biased (70%, 80%, 90%) cases, under MNIST variants with difficulties of 0.0001. As listed in the table, all models have similar accuracy to the vanilla case. This means that, re-

| Benchmark | 10% (Unbiased) | | 70% | | 80% | | 90% | |
|---|---|---|---|---|---|---|---|---|
| Test type | $M$ | $m$ | $M$ | $m$ | $M$ | $m$ | $M$ | $m$ |
| Vanilla | 99.71 | 98.60 | 99.81 | 97.96 | 100.0 | 95.92 | 99.21 | 99.29 |
| LfF | 100.0 | 97.85 | 99.61 | 98.67 | 99.81 | 98.21 | 99.21 | 99.15 |
| ReBias | 99.90 | 98.47 | 99.81 | 97.90 | 99.90 | 96.37 | 99.31 | 99.24 |
| REPAIR | 99.71 | 98.72 | 99.81 | 98.17 | 100.0 | 97.08 | 99.70 | 99.31 |
| Ours | 98.45 | 97.38 | 99.22 | 98.40 | 99.71 | 98.04 | 98.42 | 98.09 |

Table 13: Unbiased Colored MNIST accuracy results.

gardless of whether the training set is biased, all models, including ours, do not suffer from severe side effects. It shows that our algorithm can be applied without performance loss. In short, our algorithm needs not to utilize human knowledge about bias in the training set, an on/off knowledge.

**Training Time.** To see the drawbacks of training time for de-biasing, we measure the training time of 6 related works on WM case. As in Table 14, the proposed model spends $\times 1.45$ time compared to RUBi which is the fastest de-biasing method, while $\times 0.72$ faster than the other re-sampling algorithm, REPAIR.

| Vanilla | RUBi | Mixin |
|---------|------|-------|
| 12' 26" | 26' 13" | 29' 22" |
| LfF | REPAIR | Ours |
| 27' 15" | 53' 08" | 38' 00" |

Table 14: Training time

**Other metrics.** This study shows other metrics-based resampling results, (e.g., softmax response, loss, and logit value), based on sampling probability. These results show empirical justification of our gradient-based scores for de-biasing. All scores are defined to make the minority samples (unfamiliar samples) have a higher sampling probability. Each score is defined as follows:

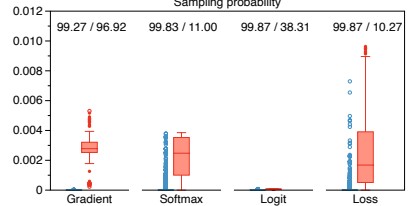

$$S_{\text{softmax}}(i) = \frac{1 - S(f(x_i, y_i))}{\sum_j (1 - S(f(x_j, y_j)))}, S_{\text{logit}}(i) = \frac{m - f(x_i, y_i)}{\sum_j (m - (f(x_j, y_j)))}, S_{\text{loss}}(i) = \frac{\mathcal{L}(x_i, y_i)}{\sum_j \mathcal{L}(x_j, y_j)},$$

where $S(\cdot)$ is softmax response and $m = \max_{k \in D}(f(x_k, y_k))$. As depicted in Figure H, our gradient-based approach has the highest performance on the average accuracy of majority and minority results. Moreover, the gradient method has a short error bar than the others, which works reasonably (softmax and loss case).

**Impact of GCE.** We checked the proposed method without GCE (trained with CE loss) under $99.5\%$ WM case. The proposed method w/o GCE achieved $(M/m)$ $99.15\%$ / $81.72\%$, which outperforms the others, while lower than the proposed method without GCE case $(99.35\%$ / $83.10\%)$.

| Vanilla | RUBi |
|---------|------|
| 98.84 / 94.78 | 98.84 / 94.66 |
| REPAIR | Ours |
| 98.84 / 95.56 | **98.55 / 96.30** |

Table 15: $|m|/|D| = 0.7$

# I  DE-NOISING

In recent noisy label cases, there have been two major trends: (1) constructing robust objective functions or regularizers (Cao et al., 2020; Yi & Wu, 2019), and (2) cleaning the noisy data (Li et al., 2019; Kim et al., 2021b). Those strategies are mostly used to reduce the influence of rare

| | Clean/Major | Clean/Minor | Noisy/Major | Noisy/Minor |
|---|---|---|---|---|
| Before denoising | 50586 | 568 | 2820 | 26 |
| FINE | 47455 | 12 | 0 | 0 |
| Ours | 50577 | 331 | 1 | 4 |

Table 16: Comparison between the state-of-the-art noisy label algorithm and the proposed de-noising module in terms of the number of preserved samples after cleansing.

samples, such as out-of-cluster samples in (Kim et al., 2021a). However, in the case of dataset bias problem, minority samples can be omitted when rare samples are discarded, resulting in de-biasing algorithms' performance degrading. As a result, erasing noisy label samples while maintaining minority samples is crucial in the dataset bias problem with noisy labels to preserve the de-biasing performance. Table 16 shows the number of preserved samples after cleansing steps. We compare the proposed de-noising module to the state-of-the-art method, FINE. It cutting-out noisy labels by using eigenvectors. As shown in Table 16, while the proposed de-noising module does not perfectly eliminate noisy samples, it does preserve clean minority samples, whereas FINE does not.

