# OpenReview forum: "Mitigating Dataset Bias Using Per-Sample Gradients From A Biased Classifier"
_ICLR.cc/2022/Conference — ICLR 2022 Submitted_

### Official Review · Reviewer_sFXQ · 2021-11-01

**Correctness:** 3
**Technical Novelty And Significance:** 3
**Empirical Novelty And Significance:** 2
**Recommendation:** 5
**Confidence:** 4

**Main Review:**

## Pros
- This work studies an important problem in machine learning. The proposed method has the benefit of not requiring supervision of biased attributes, which makes it potentially scalable to large-scale datasets with bias.
- Extensive experimental results are reported, where the proposed method showed superior performance to prior state-of-the-art on the both synthetic benchmarks and real-world datasets. It is particular interesting that all baseline methods are vulnerable to label noise and that the denoising module improves results universally.
- The writing of the paper is easy to follow. Implementation details are presented with full transparency which contributes to the reproducibility of the work.

## Cons / Questions
- The key assumption which motivates the proposed method, namely that minority samples have different gradient distributions than majority ones, deserves a more rigorous validation. For example, in addition to the aggregate measure in Figure 4b, it would be more convincing to compare histograms of per-sample gradient norms $||\nabla\theta_i||$ for both groups of training data, to ensure that the difference between the two groups is not the consequence of a few outliers.
- Despite its positive performance under synthetic settings, the significance of the proposed label denoising module is not validated on real-world datasets. In addition, the paper makes no reference to the literature on training with noisy labels (some recent work includes [A, B, C]). It is unclear how the proposed method compares to these approaches both gconceptually and in empirical results.
- Experimental results on real-world datasets are relatively weaker compared to synthetic ones. For example, the reimplemented LfF model reports significantly lower accuracy on BAR dataset (57.35% vs. 62.98%). Can the authors comment on factors that may contribute to this discrepancy? For ImageNet experiments, it would be cool if similar improvements can be observed on ImageNet-C [D] and Stylized-ImageNet [E] test sets. Finally, among the six baseline debiasing methods used in this work, only LfF and REPAIR are compared against using real-world datasets. It would be helpful if explanations are provided on the choice of baselines for these tasks.
- Results in Table 1 are presented in a slightly misleading way, as it seems to suggest that the proposed method achieves highest accuracy on both majority and minority groups. I believe the clarity can be improved by additional columns for the average of major and minor accuracies and making bold only the best numbers in each column.

## Minor comments
- Table 1 caption seems to have grammatical error: "...bolded for the on average of the majority and minority tests".
- In table 3, column titles `CelebA` and `IN/IN-A` should be swapped.

[A] Yu, Xingrui, et al. "How does disagreement help generalization against label corruption?." *International Conference on Machine Learning*. PMLR, 2019.

[B] Yi, Kun, and Jianxin Wu. "Probabilistic end-to-end noise correction for learning with noisy labels." *Proceedings of the IEEE/CVF Conference on Computer Vision and Pattern Recognition*. 2019.

[C] Li, Junnan, Richard Socher, and Steven CH Hoi. "Dividemix: Learning with noisy labels as semi-supervised learning." *arXiv preprint arXiv:2002.07394* (2020).

[D] Hendrycks, Dan, and Thomas Dietterich. "Benchmarking neural network robustness to common corruptions and perturbations." *arXiv preprint arXiv:1903.12261* (2019).

[E] Geirhos, Robert, et al. "ImageNet-trained CNNs are biased towards texture; increasing shape bias improves accuracy and robustness." *arXiv preprint arXiv:1811.12231* (2018).

**Summary Of The Paper:**

The paper proposes a de-biasing method for neural network without requiring explicit annotations of biases. The authors propose a resampling-based approach that automatically detects underrepresented samples based on the direction and magnitude of their loss gradients, then balances the dataset by rejection sampling. The paper further proposes a denoising module to identify and filter out noisy samples under the presence of label noise. The gradient-based de-biasing method is empirically shown to outperform prior work on both controlled synthetic and real-world datasets.

**Summary Of The Review:**

The paper presents a novel and simple debiasing approach for neural networks, as well as a denoising module for training with label noise. The proposed debiasing method shows promising results especially on synthetic benchmarks, comparing against various state-of-the-art algorithms; however, I believe the concerns listed in the section above needs to be addressed to fully justify the contributions of the work. I am setting my initial score to borderline reject but would be happy to update it based on the reviewers' response.

---

> ### Author Response · Authors · 2021-11-14
> **Response to reviewer sFXQ (2/3)**
>
> __Q3__) Experimental results on real-world datasets are relatively weaker compared to synthetic ones. For example, the reimplemented LfF model reports significantly lower accuracy on BAR dataset (57.35% vs. 62.98%). Can the authors comment on factors that may contribute to this discrepancy?
>
> __A3__) As far as we know, almost all prior works optimized hyperparameters using the test dataset (by viewing the offered official code), which we argue that this is impractical. We reimplement by splitting the training dataset for validation, which could lead to inconsistencies in performance.
>
> We strongly believe that hyperparameters must be tuned via the validation dataset rather than the test set. However, in this dataset bias problem, it is difficult to know whether the training set is biased or not.  It is an obstacle to tune for the performance of the minority samples. To overcome this, we believe that the target model must be robust to hyperparameter sensitivity. As in the Hyperparameter sensitivity paragraph in Appendix H, our algorithm is less sensitive than the other algorithms.
>
> ---
>
> __Q4__) Finally, among the six baseline debiasing methods used in this work, only LfF and REPAIR are compared against using real-world datasets. It would be helpful if explanations are provided on the choice of baselines for these tasks.
>
> __A4__) We chose two algorithms: the most recent re-sampling algorithm and the approach for changing objective methods. When adjusting the objective, LfF is the most recent version, and REPAIR is the most recent in re-sampling.
>
> ---
>
> __Q5__) Results in Table 1 are presented in a slightly misleading way, as it seems to suggest that the proposed method achieves highest accuracy on both majority and minority groups. I believe the clarity can be improved by additional columns for the average of major and minor accuracies and making bold only the best numbers in each column.
>
> __A5__) Table 1 has been updated to highlight the best performance on minority samples what de-biasing algorithms mainly focusing on. Thank you for your clear illustration remark.
>
> ---
>
> __Q6__) Minor comments
>
> __A6__) Thank you for your minor comments, we will revise the paper following your recommendations.

---

> ### Author Response · Authors · 2021-11-14
> **Response to reviewer sFXQ (2/3)**
>
> __Q2__) Despite its positive performance under synthetic settings, the significance of the proposed label denoising module is not validated on real-world datasets. In addition, the paper makes no reference to the literature on training with noisy labels (some recent work includes [A, B, C]). It is unclear how the proposed method compares to these approaches both conceptually and in empirical results.
>
> __A2__) To the best of our knowledge, our de-noising module is the first to observe the case where noisy labels are included in the dataset bias problem. It is difficult to discriminate between minority samples and noisy labels in a biased dataset. Moreover, the noisy-biased dataset has an important constraint to design a de-noising method such that the algorithm should keep as many of the minority samples as possible since the number of discarded minority samples affects the de-biasing algorithm.
>
> There have been two key trends in recent noisy label cases: (1) developing robust objective functions or regularizations [1,2] and (2) cleaning the noisy data [3,4]. Here, we consider cleansing noisy data. We will report the ratio of discarding minority samples from ours and one prior cleansing-based noisy label approach [4] using their official code.
> Colored MNIST case (99%)
> C-M: Clean Major / C-m: Clean minor / N-M: Noisy Major / N-m: Noisy minor
>
> |Type|C-M|C-m|N-M|N-m|
> |:---:|:---:|:---:|:---:|:---:|
> |Before deno.	|	50586	|	568	|	2820	|	26	|
> |Ours		|	50577	|	__331__	|	1	|	4	|
> |FINE		|	47455	|	__12__	|	0	|	0	|
>
>
> Before describing the above results, FINE selects clean samples for every epoch, but to ensure a fair comparison with our de-noising module, we reported the first epoch result. The proposed  de-noising module saved minority samples while few samples were contained. On the other hand, FINE not only removes clearly noisy samples, but also minority samples. We believe that the proposed method can remove noisy samples while successfully saving minority samples.
>
> [1] Heteroskedastic and imbalanced deep learning with adaptive regularization ICLR2021
> [2] Probabilistic end-to-end noise correction for learning with noisy labels CVPR2019
> [3] How does disagreement help generalization against label corruption? PMLR2019
> [4] FINE samples for learning with Noisy Labels  arXiv2021.02.11628 (NeurIPS2021)

---

> ### Author Response · Authors · 2021-11-14
> **Response to reviewer sFXQ (1/3)**
>
> We are grateful for your time and constructive feedback; we appreciate all of your thoughtful feedback. (1) Statistics of sampling probability based on the magnitude. (2) Comparison with other noisy label methods. (3) Results with the BAR benchmark. (4) Results for other cases, i.e., ImageNet-C, Stylized ImageNet. (5) Reason for choosing specific baselines for real-world benchmarks. Each of your concerns is addressed individually.
>
> ---
>
> __Q1__) The key assumption which motivates the proposed method, namely that minority samples have different gradient distributions than majority ones, deserves a more rigorous validation. For example, in addition to the aggregate measure in Figure 4b, it would be more convincing to compare histograms of per-sample gradient norms ||∇θi|| for both groups of training data, to ensure that the difference between the two groups is not the consequence of a few outliers.
>
> __A1__) We described a histogram of per-sample gradients statistics in the manuscript. Here are the histogram examples of the sampling probability in two cases {C1: CM 99.5\%, C2: CM 99\%}. All rows are normalized by their max value. As in the table, minority samples that have a higher sampling probability have higher portions than the majority, which means that our per-sample gradient-based splitting does not select very few outliers.
> Additionally, to convey our intuition precisely in the case of DNNs, we show a number/portion of samples at each bin. N and V represent normalized Euclidean norm (N) and  vMF likelihood (V). We conducted this in two cases {C1: CM 99\%, C2: CM 99.5\%}.
>
>
> ### Portion
> |Type	| 0-0.1 	| 0.1-0.2 	|0.2-0.3 	| 0.3-0.4 	| 0.4-0.5 	| 0.5-0.6 	| 0.6-0.7	| 0.7-0.8 	| 0.8-0.9 	| 0.9-1.0 |
> |:---:|:---:|:---:|:---:|:---:|:---:|:---:|:---:|:---:|:---:|:---:|
> |C1-N/M	|99.86	|__67.85__	|__30.34__	|__8.96__	|__4.49__	|__13.07__	|__18.42__	|__20.00__	|__33.33__	|__0.00__|
> |C1-N/m	|0.14	|__32.15__	|__69.66__	|__91.04__	|__95.51__	|__86.92__	|__81.58__	|__80.00__	|__66.67__	|__100.0__|
> |C1-V/M	|__27.92__	|-	|-	|-	|-	|-	|-|	99.42	|99.50	|99.65	|
> |C1-V/m	|__72.08__	|-	|-	|-	|-	|-	|-|	0.58	|0.50	|0.35	|
>
> |Type	| 0-0.1 	| 0.1-0.2 	|0.2-0.3 	| 0.3-0.4 	| 0.4-0.5 	| 0.5-0.6 	| 0.6-0.7	| 0.7-0.8 	| 0.8-0.9 	| 0.9-1.0 |
> |:---:|:---:|:---:|:---:|:---:|:---:|:---:|:---:|:---:|:---:|:---:|
> |C2-N/M	|99.96	|99.16	|96.82	|__65.62__	|__45.38__	|__32.84__	|__49.38__	|__69.77__	|__72.22__	|__66.67__|
> |C2-N/m	|0.04	|0.83	|3.17	|__34.38__	|__54.62__	|__67.18__	|__50.61__	|__30.23__	|__27.78__	|__33.33__|
> |C2-V/M	|__89.20__	|-	|-	|100	|100	|-	|-	|99.69	|99.64	|99.70|
> |C2-V/m	|__10.80__	|-	|-	|0	|0	|-	|-	|0.31	|0.36	|0.30|
>
>
> ### \# of samples
> |Type	| 0-0.1 	| 0.1-0.2 	|0.2-0.3 	| 0.3-0.4 	| 0.4-0.5 	| 0.5-0.6 	| 0.6-0.7	| 0.7-0.8 	| 0.8-0.9 	| 0.9-1.0 |
> |:---:|:---:|:---:|:---:|:---:|:---:|:---:|:---:|:---:|:---:|:---:|
> |C1-N/M	|53275	|__57__	|__27__	|__12__	|__7__	|__17__	|__7__	|__2__	|__2__	|__0__|
> |C1-N/m	|76	|__27__	|__62__	|__122__	|__149__	|__113__	|__31__	|__8__	|__4__	|__2__|
> |C1-V/M	|__146__	|0	|0	|0	|0	|0	|0	|6040	|10922	|36298|
> |C1-V/m	|__377__	|0	|0	|0	|0	|0	|0	|35	|55	|127|
>
> |Type	| 0-0.1 	| 0.1-0.2 	|0.2-0.3 	| 0.3-0.4 	| 0.4-0.5 	| 0.5-0.6 	| 0.6-0.7	| 0.7-0.8 	| 0.8-0.9 	| 0.9-1.0 |
> |:---:|:---:|:---:|:---:|:---:|:---:|:---:|:---:|:---:|:---:|:---:|
> |C2-N/M	|53107	|237	|122	|__63__	|__59__	|__43__	|__40__	|__30__	|__13__	|__4__|
> |C2-N/m	|23	|2	|4	|__33__	|__71__	|__88__	|__41__	|__13__	|__5__	|__2__|
> |C2-V/M	|__834__	|0	|0	|2	|2	|0	|0	|6083	|37475	|9322|
> |C2-V/m	|__101__	|0	|0	|0	|0	|0	|0	|19	|134	|28|
>
> As shown in the Table, the portion of minority samples increases when magnitude increases, and directional likelihood decreases. We emphasized that the ratio of minority is 10% or more as a bold font.  From the convex example to the DNN-based empirical results, we believe that the above results empirically support our intuition.

---

> ### Comment · Reviewer_sFXQ · 2021-11-28
> **Response to Authors' Comments**
>
> I thank the authors for the very detailed response. The distributions of loss norm and directions are much appreciated, though their clarity can be improved by 1) plotting histograms of majority/minority classes individually and overlap on the same figure, and 2) replacing bin index with norm/vMF values. It would be ideal if the experiment were repeated on all synthetic datasets (Watermarked MNIST, Cartoon), and if a dedicated paragraph were included to discuss the results in detail, e.g. why the percentage of minority samples begins to drop after bin #6 in the 99.5% setting. Comparison to FINE is quite interesting and better motivates the need to study label noise in the context of dataset bias. However, I would still love to see ablation studies on real datasets (e.g. compare to results without denoising, or denoised using FINE) to fully justify the practical importance of proposed module.

---

### Official Review · Reviewer_BR78 · 2021-11-02

**Correctness:** 3
**Technical Novelty And Significance:** 3
**Empirical Novelty And Significance:** 2
**Recommendation:** 5
**Confidence:** 4

**Main Review:**

The proposed idea of using gradient-based scoring for de-biasing seems novel and effective. I have the following concerns and questions.

This work is based on the hypothesis that the “gradients have remarkable differences between samples generating prejudice and the others”. By observing that the label noise can degrade the performance of the proposed gradient-based model, they added the de-noising module to alleviate this issue. However, I’m not convinced if the noisy label is the only critical factor that can negatively affect the performance when using the proposed gradient-based scheme. I wonder what other factors (e.g., adversarial attacks) can degrade the performance and how to tackle them. Can you add more analysis and discussion on this?

Figure 4 is drawn using the Colored MNIST. This result may be useful to illustrate the proposed hypothesis, but it may not be enough to show the generality because the dataset has a very simple structure and variation. I wonder how the plots for more natural datasets would look like, and if any theoretical analysis can be included.

The model assumes that the biased samples make up the majority of the training set. The experiments are also based on this assumption, and the proportions of the unbiased (minority) samples in the experiments are 5% or less in all the experiments. I wonder how realistic this scenario is and how the proposed method performs on a dataset that contains a higher portion of unbiased (minority) training data (e.g., 10%, 20%, 30%,…).

In Table 2, comparing the accuracies for the Minor group, which I think is more important in the performance comparison, the performance of the proposed model “with de-noising”, compared to the case without it, is improved at the 99%/10% setting, but not for the remaining cases of 99%/5% and 99%/0% (except for Watermarked MNIST, 99%/5%, showing very low accuracy). Therefore, this result is not sufficient to validate the proposed de-noising idea. More experiments and/or analyses should be provided.

Minor comments:

- Figure 3: |m|/|D| -> |M|/|D|
- Page 6, Section 4.2: “From the harmonic-mean property has a higher value” (need to revise)
- In Table 2 - 99%/5% - Without de-noising – Major M:
-- Colored MNIST: LfF performance is written as 00.00. Should it be 100.00?
-- Watermarked MNIST: the performance from “Ours” is 33.26, which is much lower than the other methods (over 90). Is it a typo or any reason for this?
- In Table 1, the method names of “Proposed” and “Ours” are both used. It’s better to make it consistent.
- Page 9, Section 6.2
-- ... an accuracy similar to that o the major case. -> … an accuracy similar to that of the major case.
-- de-basing -> de-biasing


**Summary Of The Paper:**

This paper proposes a gradient-based resampling scheme for de-biasing. The authors construct two types of scores that leverage gradients of the biased model: the magnitude and the direction of the gradients. In addition, to mitigate the side effects from the noisy labels, the authors also propose a de-noising module, which can be easily applied to any de-biasing algorithms. Extensive experimentation validates the proposed method against other state-of-the-art methods.

**Summary Of The Review:**

This paper present a novel idea for de-biasing and its performance is validated on limited scenarios. More validation and analysis of the used hypothesis and the model would strengthen the paper.

---

> ### Author Response · Authors · 2021-11-14
> **Response to reviewer BR78 (3/3)**
>
> __Q3__) The model assumes that the biased samples make up the majority of the training set. The experiments are also based on this assumption, and the proportions of the unbiased (minority) samples in the experiments are 5% or less in all the experiments. I wonder how realistic this scenario is and how the proposed method performs on a dataset that contains a higher portion of unbiased (minority) training data (e.g., 10%, 20%, 30%,…).
>
> __A3__) First, we believe that when the training set is not biased, de-biasing algorithms must show similar performance to the vanilla method. To demonstrate this, we ran MNIST variant benchmarks with a higher portion of minority and unbiased cases (i.e., |m|/|D|= \{90\% (Unbiased), 30\%, 20\%, 10\%\}). Note that the 10% case is an unbiased case because all colors are uniformly distributed in each class. For all cases, our and other methods produced equivalent results to the vanilla method. In our method, this finding is due to the fact that the sampling probability decreases as the minority’s portion decreases. The following tables show the results. Furthermore, all sampling probabilities of majority and minority sets are also reported.
>
> Rarely biased, Unbiased cases (CM)
>
> |	Type		| 10%(unbiased)|	70%	| 80% |90%|
> |:---:|:---:|:---:|:---:|:---:|
> |	Vanilla		|	99.21%/99.29%	|	99.71%/98.60%	|	99.81%/97.96%	|	100.0%/95.92%	|
> |	LFF		|	99.21%/99.15%	|	100.0%/97.85%	|	99.61%/98.67%	|	99.81%/98.21%	|
> |	Rebias		|	99.31%/99.24%	|	99.90%/98.47%	|	99.81%/97.90%	|	99.90%/96.37%	|
> |	REPAIR	|	99.70%/99.31%	|	99.71%/98.72%	|	99.81%/98.17%	|	100.0%/97.08%	|
> |	Ours		|	98.42%/98.09%	|	98.45%/97.38%	|	99.22%/98.40%	|	99.71%/98.04%	|
>
> ---
> __Q4__) In Table 2, comparing the accuracies for the Minor group, which I think is more important in the performance comparison, the performance of the proposed model “with de-noising”, compared to the case without it, is improved at the 99%/10% setting, but not for the remaining cases of 99%/5% and 99%/0% (except for Watermarked MNIST, 99%/5%, showing very low accuracy). Therefore, this result is not sufficient to validate the proposed de-noising idea. More experiments and/or analyses should be provided.
>
> __A4__) First of all, we have corrected the error in Table 2. Please see the revised manuscript and comment for all reviewers and AC. We apologize for causing confusion. The performance of the w/o de-noising performance on the average of majority and minority sets declined as the noisy ratio grew, as shown in the revised Table 2; for example, in the WM situation, 99.67/91.98 -> 52.37/45.54 -> 28.48/26.11. However, after employing the de-noising module in all circumstances, its performance is comparable to the without noisy label (0\%) case. Furthermore, the performance suffers when the  de-noising module eliminates minority samples. As shown in Figure 6 in the manuscript, distinguishing between noisy labels and minority samples is challenging, resulting in a performance drop in the minority set.
>
> ---
>
> __Q5__) Minor comments
>
> __A5__) Thank you for your minor comments; we will revise the paper following your recommendations.

---

> ### Author Response · Authors · 2021-11-14
> **Response to reviewer BR78 (2/3)**
>
> __Q2__) Figure 4 is drawn using the Colored MNIST. This result may be useful to illustrate the proposed hypothesis, but it may not be enough to show the generality because the dataset has a very simple structure and variation. I wonder how the plots for more natural datasets would look like, and if any theoretical analysis can be included.
>
> __A2__) To convey our simple motivation for the relationship between the per-sample gradient and the majority/minority splits, we first describe a simple convex example and then describe our per-samples histogram results of sampling probability.
>
> Starting with a simple convex example, let us assume that there are $P$ numbers of samples with loss functions are $(x)^2$ and $Q$ samples whose functions are $(x-10)^2$. These types have optimal points $x = 0$ and $x = 10$, respectively. The aggregated loss function is $P \times (x)^2 + Q \times (x-10)^2$, and the optimizer converges to the point $x’ = 10/(P+Q)$. At the optimal point x’, the gradients of each sample in each type are $20/(P+Q)$ and $20/(P+Q)-20$. When$ P:Q = 99:1 $(99% bias), the gradients are $0.2$ and $-19.8$, which differ in terms of direction and magnitude. We hypothesize that majority samples and minority samples define different loss functions as in the simple example. Then, at the optimal point, the majority will have a shorter gradient vector with an opposite direction compared to the minority.
>
>
> Additionally, to convey our intuition precisely in the case of DNNs, we show a number/portion of samples at each bin. N and V represent normalized Euclidean norm (N) and  vMF likelihood (V). We conducted this in two cases {C1: CM 99\%, C2: CM 99.5\%}.
>
>
> ### Portion
> |Type	| 0-0.1 	| 0.1-0.2 	|0.2-0.3 	| 0.3-0.4 	| 0.4-0.5 	| 0.5-0.6 	| 0.6-0.7	| 0.7-0.8 	| 0.8-0.9 	| 0.9-1.0 |
> |:---:|:---:|:---:|:---:|:---:|:---:|:---:|:---:|:---:|:---:|:---:|
> |C1-N/M	|99.86	|__67.85__	|__30.34__	|__8.96__	|__4.49__	|__13.07__	|__18.42__	|__20.00__	|__33.33__	|__0.00__|
> |C1-N/m	|0.14	|__32.15__	|__69.66__	|__91.04__	|__95.51__	|__86.92__	|__81.58__	|__80.00__	|__66.67__	|__100.0__|
> |C1-V/M	|__27.92__	|-	|-	|-	|-	|-	|-|	99.42	|99.50	|99.65	|
> |C1-V/m	|__72.08__	|-	|-	|-	|-	|-	|-|	0.58	|0.50	|0.35	|
>
> |Type	| 0-0.1 	| 0.1-0.2 	|0.2-0.3 	| 0.3-0.4 	| 0.4-0.5 	| 0.5-0.6 	| 0.6-0.7	| 0.7-0.8 	| 0.8-0.9 	| 0.9-1.0 |
> |:---:|:---:|:---:|:---:|:---:|:---:|:---:|:---:|:---:|:---:|:---:|
> |C2-N/M	|99.96	|99.16	|96.82	|__65.62__	|__45.38__	|__32.84__	|__49.38__	|__69.77__	|__72.22__	|__66.67__|
> |C2-N/m	|0.04	|0.83	|3.17	|__34.38__	|__54.62__	|__67.18__	|__50.61__	|__30.23__	|__27.78__	|__33.33__|
> |C2-V/M	|__89.20__	|-	|-	|100	|100	|-	|-	|99.69	|99.64	|99.70|
> |C2-V/m	|__10.80__	|-	|-	|0	|0	|-	|-	|0.31	|0.36	|0.30|
>
>
> ### \# of samples
> |Type	| 0-0.1 	| 0.1-0.2 	|0.2-0.3 	| 0.3-0.4 	| 0.4-0.5 	| 0.5-0.6 	| 0.6-0.7	| 0.7-0.8 	| 0.8-0.9 	| 0.9-1.0 |
> |:---:|:---:|:---:|:---:|:---:|:---:|:---:|:---:|:---:|:---:|:---:|
> |C1-N/M	|53275	|__57__	|__27__	|__12__	|__7__	|__17__	|__7__	|__2__	|__2__	|__0__|
> |C1-N/m	|76	|__27__	|__62__	|__122__	|__149__	|__113__	|__31__	|__8__	|__4__	|__2__|
> |C1-V/M	|__146__	|0	|0	|0	|0	|0	|0	|6040	|10922	|36298|
> |C1-V/m	|__377__	|0	|0	|0	|0	|0	|0	|35	|55	|127|
>
> |Type	| 0-0.1 	| 0.1-0.2 	|0.2-0.3 	| 0.3-0.4 	| 0.4-0.5 	| 0.5-0.6 	| 0.6-0.7	| 0.7-0.8 	| 0.8-0.9 	| 0.9-1.0 |
> |:---:|:---:|:---:|:---:|:---:|:---:|:---:|:---:|:---:|:---:|:---:|
> |C2-N/M	|53107	|237	|122	|__63__	|__59__	|__43__	|__40__	|__30__	|__13__	|__4__|
> |C2-N/m	|23	|2	|4	|__33__	|__71__	|__88__	|__41__	|__13__	|__5__	|__2__|
> |C2-V/M	|__834__	|0	|0	|2	|2	|0	|0	|6083	|37475	|9322|
> |C2-V/m	|__101__	|0	|0	|0	|0	|0	|0	|19	|134	|28|
>
> As shown in the Table, the portion of minority samples increases when magnitude increases, and directional likelihood decreases. We emphasized that the ratio of minority is 10% or more as a bold font.  From the convex example to the DNN-based empirical results, we believe that the above results empirically support our intuition.

---

> ### Author Response · Authors · 2021-11-14
> **Response to reviewer BR78 (1/3)**
>
> We are grateful for your time and constructive feedback; we appreciate all of your thoughtful feedback: (1) other kinds of vulnerabilities on ours and the de-biasing algorithms   (2) more explanation of our motivation, (3) results of rare biased cases, (4) explanation of Table 2 in the manuscript. Each of your concerns is addressed individually.
>
> ---
> __Q1__) I’m not convinced if the noisy label is the only critical factor that can negatively affect the performance when using the proposed gradient-based scheme. I wonder what other factors (e.g., adversarial attacks) can degrade the performance and how to tackle them. Can you add more analysis and discussion on this?
>
> __A1__) As far as we know, this work is the first paper to consider other factors in de-biasing algorithms. Among various factors that can affect the de-biasing algorithm, we first look at the noisy label problem for two reasons: (1) It is one of the most common noisy training set issues. (2) The methodology of noisy labels is in conflict with the de-biasing method, which  means that both must be considered carefully. To be more specific about the second reason, the usual solutions of the noisy labels basically rely on mitigating the impact of the samples, which is difficult to guess (i.e., noisy label) [1]. On the other hand, in de-biasing methods, hard-guess samples (i.e., minority) must be highlighted. Thank you for your constructive opinion, and we agree with your comment, other factors can affect this, but we think that this can be addressed in future work in this de-biasing research.

---

### Official Review · Reviewer_3HUA · 2021-11-02

**Correctness:** 3
**Technical Novelty And Significance:** 2
**Empirical Novelty And Significance:** 3
**Recommendation:** 6
**Confidence:** 4

**Main Review:**

The proposed method differs from prior work in the novel scoring function. However, the proposed scoring function is not well-motivated. Despite the success of better performances, it is hard to be convinced that the per-sample gradients are able to differentiate between the samples drawn from majority or minority group. The authors show only 4 examples in Color-MNIST in figure 4, which are not representative enough.

**Strength**
- Extensive experimental results across different setting and different baselines.
- Strong performance compared to the baselines

**Weakness**
- Lack of motivation and analysis on the connections between per-sample gradients and the majority/minority splits (in more complex datasets)

**Others**
- For table 3, I would be interested to see the accuracy of the minority as well since it help validate if the proposed method works under real-world correlation.

**Summary Of The Paper:**

The paper propose to use the per-sample gradients from the biased classifier to tackle the dataset bias. Following prior work, this work adopts a three-step approach: 1) train a classifier on biased dataset 2) identify which samples are from the minority and which are not 3) train a classifier with re-sampling. The proposed per-sample gradient-based scores are used in step 2. The motivation is purely from the empirical observation that samples from the majority group tend to have similar gradient magnitude and direction. From the observation, they derive a scoring function for re-sampling. The proposed method work pretty well empirically compared to various baselines.

Aside from the novel scoring function, since the noisy labeled data might be accidentally considered as minority group, the paper introduces a de-noising step to tackle the potential noise in the dataset. The de-noising step further improves the performances.

**Summary Of The Review:**

The paper shows good empirical results on mitigating dataset bias by devising a novel scoring function for re-sampling. The scoring function is the key in this work, but is not well-motivated enough. The paper can gains a lot of values by providing more analysis on the connections between per-sample gradients and the majority/minority sets.


---

Updated on Dec. 6.
Though the paper still lacks of theoretic motivations, it does provide convincing empirical findings. I raised my score from 5 to 6.

---

> ### Author Response · Authors · 2021-11-14
> **Response to reviewer 3HUA (2/2)**
>
> __Q2__) For table 3, I would be interested to see the accuracy of the minority as well since it helps validate if the proposed method works under real-world correlation.
>
> __A2__) For the CelebA benchmark, we included both the majority and minority results. Due to the lack of explicit information about minorities, we were unable to describe minority results for both ImageNet/ImageNet-A and BAR. As shown in the Table below (see also in Appendix F), our algorithm performs better on minority samples. We believe that our algorithm can balance between the two $M$ and $m$ sets.
>
> |Algorithm 	|	Majority	|	Minority	|	Average	|
> |:---:|:---:|:---:|:---:|
> |Vanilla		|	93.16%	|	71.56%	|	82.36%	|
> |LfF		|	92.60%	|	69.67%	|	81.13%	|
> |REPAIR	|	92.54%	|	73.48%	|	83.01%	|
> |Ours		|	88.18%	|	83.83%	|	86.01%	|
> ----------------------------------------------------------------------------------------------------------

---

> ### Author Response · Authors · 2021-11-14
> **Response to reviewer 3HUA (1/2)**
>
> We are grateful for your time and constructive feedback; we appreciate all of your thoughtful feedback. (1) Additional explanation about connections between per-sample gradients and the majority/minority splits and (2) Report the accuracy of  majority/minority under real-world correlation. Each of your concerns is addressed individually.
>
> __Q1__) Lack of motivation and analysis on the connections between per-sample gradients and the majority/minority splits (in more complex datasets).
>
> __A1__) To convey our simple motivation for the relationship between the per-sample gradient and majority/minority splits, we first describe a simple convex example and then our per-sample histogram results of sampling probability.
>
> Starting with a simple convex example, let us assume that there are $P$ numbers of samples with loss functions are $(x)^2$ and $Q$ samples whose functions are $(x-10)^2$. These types have optimal points $x = 0$ and $x = 10$, respectively. The aggregated loss function is $P \times (x)^2 + Q \times (x-10)^2$, and the optimizer converges to the point $x’ = 10/(P+Q)$. At the optimal point $x’$, the gradients of each sample in each type are $20/(P+Q)$ and $20/(P+Q)-20$. When $P:Q = 99:1$ (99% bias), the gradients are $0.2$ and $-19.8$, which differ in terms of direction and magnitude. We hypothesize that majority samples and minority samples define different loss functions as in the simple example. Then, at the optimal point, the majority will have a shorter gradient vector with an opposite direction compared to the minority.
>
> Additionally, to convey our intuition precisely in the case of DNNs, we show a number/portion of samples at each bin. N and V represent normalized Euclidean norm (N) and  vMF likelihood (V). We conducted this in two cases {C1: CM 99\%, C2: CM 99.5\%}.
>
>
> ### Portion
> |Type	| 0-0.1 	| 0.1-0.2 	|0.2-0.3 	| 0.3-0.4 	| 0.4-0.5 	| 0.5-0.6 	| 0.6-0.7	| 0.7-0.8 	| 0.8-0.9 	| 0.9-1.0 |
> |:---:|:---:|:---:|:---:|:---:|:---:|:---:|:---:|:---:|:---:|:---:|
> |C1-N/M	|99.86	|__67.85__	|__30.34__	|__8.96__	|__4.49__	|__13.07__	|__18.42__	|__20.00__	|__33.33__	|__0.00__|
> |C1-N/m	|0.14	|__32.15__	|__69.66__	|__91.04__	|__95.51__	|__86.92__	|__81.58__	|__80.00__	|__66.67__	|__100.0__|
> |C1-V/M	|__27.92__	|-	|-	|-	|-	|-	|-|	99.42	|99.50	|99.65	|
> |C1-V/m	|__72.08__	|-	|-	|-	|-	|-	|-|	0.58	|0.50	|0.35	|
>
> |Type	| 0-0.1 	| 0.1-0.2 	|0.2-0.3 	| 0.3-0.4 	| 0.4-0.5 	| 0.5-0.6 	| 0.6-0.7	| 0.7-0.8 	| 0.8-0.9 	| 0.9-1.0 |
> |:---:|:---:|:---:|:---:|:---:|:---:|:---:|:---:|:---:|:---:|:---:|
> |C2-N/M	|99.96	|99.16	|96.82	|__65.62__	|__45.38__	|__32.84__	|__49.38__	|__69.77__	|__72.22__	|__66.67__|
> |C2-N/m	|0.04	|0.83	|3.17	|__34.38__	|__54.62__	|__67.18__	|__50.61__	|__30.23__	|__27.78__	|__33.33__|
> |C2-V/M	|__89.20__	|-	|-	|100	|100	|-	|-	|99.69	|99.64	|99.70|
> |C2-V/m	|__10.80__	|-	|-	|0	|0	|-	|-	|0.31	|0.36	|0.30|
>
>
> ### \# of samples
> |Type	| 0-0.1 	| 0.1-0.2 	|0.2-0.3 	| 0.3-0.4 	| 0.4-0.5 	| 0.5-0.6 	| 0.6-0.7	| 0.7-0.8 	| 0.8-0.9 	| 0.9-1.0 |
> |:---:|:---:|:---:|:---:|:---:|:---:|:---:|:---:|:---:|:---:|:---:|
> |C1-N/M	|53275	|__57__|__27__	|__12__	|__7__	|__17__	|__7__	|__2__	|__2__	|__0__|
> |C1-N/m	|76	|__27__	|__62__	|__122__	|__149__	|__113__	|__31__	|__8__	|__4__	|__2__|
> |C1-V/M	|__146__	|0	|0	|0	|0	|0	|0	|6040	|10922	|36298|
> |C1-V/m	|__377__	|0	|0	|0	|0	|0	|0	|35	|55	|127|
>
> |Type	| 0-0.1 	| 0.1-0.2 	|0.2-0.3 	| 0.3-0.4 	| 0.4-0.5 	| 0.5-0.6 	| 0.6-0.7	| 0.7-0.8 	| 0.8-0.9 	| 0.9-1.0 |
> |:---:|:---:|:---:|:---:|:---:|:---:|:---:|:---:|:---:|:---:|:---:|
> |C2-N/M	|53107	|237|122|__63__	|__59__	|__43__	|__40__	|__30__	|__13__	|__4__|
> |C2-N/m	|23	|2|4|__33__	|__71__	|__88__	|__41__	|__13__	|__5__	|__2__|
> |C2-V/M	|__834__	|0	|0	|2	|2	|0	|0	|6083	|37475	|9322|
> |C2-V/m	|__101__	|0	|0	|0	|0	|0	|0	|19	|134	|28|
>
> As shown in the Table, the portion of minority samples increases when magnitude increases, and directional likelihood decreases. We emphasized that the ratio of minority is 10% or more as a bold font.  From the convex example to the DNN-based empirical results, we believe that the above results empirically support our intuition.

---

> > ### Comment · Reviewer_3HUA · 2021-11-16
> > **Histogram is much clearer than figure.4**
> >
> > Can the authors describe the row name in details? Does "C2-C" mean C2 with *clean* label? Also, I cannot find the results of *vMF likelihood*.

---

> > > ### Author Response · Authors · 2021-11-16
> > > **Notation change**
> > >
> > > Thanks for your response and sorry for making confusion. To indicate vMF likelihood, we edited our response once and changed the notation "C2-C" to "C2-V" before your mention.

---

> > > > ### Comment · Reviewer_3HUA · 2021-11-16
> > > > **Two questions regarding the histogram**
> > > >
> > > > 1. "C1: CM 99.5%, C2: CM 99%" is interpreted as "C1: Color MNIST with bias ratio 99.5: 0.5, C2: Color MNIST with bias ratio 99: 1", right? From the histogram you show, the size of minority in C2 (282) is **less** than that in C1 (594). Is there anything wrong here?
> > > > 2. Comparing C1 and C2 in the low directional likelihood bin (0~0.1), minority accounts for 72.08% in C1 but 10.80% in C2. Can the author further elaborate the potential reason of this mismatch?

---

> > > > > ### Author Response · Authors · 2021-11-16
> > > > > **Response to the questions**
> > > > >
> > > > > Thank you for your response. This is our answer for your question.
> > > > >
> > > > > 1. C1: CM 99% and C2: CM 99.5% is correct, that you noted. Therefore, the number of minority samples in 99% (282) is about twice that of 99.5% (594). (This is corrected in the preceding response.)
> > > > >
> > > > > 2. As you pointed out, the portion of the majority samples is larger than the other case (99%). However, as mentioned in the manuscript, we expected that magnitude and direction are utilized in complementarily, and aggregated score is robust to distinguish majority and minority samples from outliers. The following tables summarizes the percentage of aggregated score at each bin.
> > > > > $S = 1./ (1./M + V)$, where $M$ and $V$ represent magnitude and vMF likelihood, respectively.
> > > > >
> > > > > ---
> > > > > |Type|0-0.1|0.1-0.2|0.2-0.3|0.3-0.4|0.4-0.5|0.5-0.6|0.6-.7|0.7-0.8|0.8-0.9|0.9-1.0|
> > > > > |:---:|:---:|:---:|:---:|:---:|:---:|:---:|:---:|:---:|:---:|:---:|
> > > > > |Major|99.96|99.16|96.77|__67.35__|__46.40__|__31.85__|__48.15__|__70.45__|__72.22__|__66.67__|
> > > > > |Minor|0.04|0.84|3.23|__32.65__|__53.60__|__68.15__|__51.85__|__29.54__|__27.78__|__33.33__|

---

### Official Review · Reviewer_WQAv · 2021-11-03

**Correctness:** 3
**Technical Novelty And Significance:** 2
**Empirical Novelty And Significance:** 2
**Recommendation:** 6
**Confidence:** 4

**Main Review:**

There is a lot to like about this paper. I find the hypothesis about the "per sample" gradient can be used to identify "biased" samples interesting. The authors also correctly discuss the accompanying argument that this has close ties to label noise. The results also show a remarkable improvement in the commonly used datasets. However, I am increasingly wary of testing bias mitigation algorithms on the same toy settings, and their ability to actually quantify the degree to which these algorithms are actually working is under increasing suspicion.

Strengths:

S1.  Great results: The results for varying levels of "bias" as well as the success of the proposed "denoising" setup is remarkable for the datasets tested. The results are vastly superior to other comparable methods in three different datasets and even better for extreme levels of bias (99.5%).

S2. Interesting Hypothesis and approach: The paper proposes an interesting hypothesis about the differences between gradient magnitude and direction (as measured by its proximity to an "average" gradient direction for all samples) look different for biased as compared to "regular" data sample. This discussion would be incomplete without a discussion of its relationship with label noise which the authors rightly address. The proposed denoising module is shown to mitigate the effects of label bias on various debiasing algorithms including the proposed one. While I am not fully convinced of the hypothesis is universally valid, and whether the denoising module is truly capable of dealing with label noise per se, the empirical validation, at least in the datasets tested, is hard to refute in the context of debiasing.

Weaknesses:


W1. Unclear how bias model is implemented: In prior works, such as LearnedMixinH and Rubi for VQA, the bias-only model takes question-only model as a biased model which assumes that if an answer could be guessed only using the question, then that sample can be assumed to be biased. There isn't a straightforward parallel for this in color-MNIST. For color-MNIST, REPAIR proposes to train a classifier using only the RGB value of the input. It is unclear how 1) the biased model is trained for the proposed algorithm, 2) How were they trained for Rubi?

W2: Serious doubts about "scalability" and assumption about known bias:  There are various questions about the "scalability" of the proposed method. One common theme across these datasets is that they can be "learned" (at least the biased version) with a much smaller amount of data than is present in the training set. Hence, a rejection sampling-based method can work even when the minority set (m) diminishes. Next, the proposed algorithm falls under the category called variously as  "supervised", "known", or "explicit" bias mitigation algorithm. In other words, the algorithm assumes knowledge about which of the factors were biased so that a suitable "bias-only" model can be trained by leveraging only the "bias". Such methods can be unscalable to all but a single, well-defined type of bias. This could be addressed by demonstrating the algorithm's success in more difficult scenarios (https://arxiv.org/abs/2104.00170).

**Summary Of The Paper:**

The paper proposes a new bias-mitigation technique. The main hypothesis of the paper is that there are differences in gradients for "biased" samples (or samples that are "rare") compared to majority patterns in the training data. Using this, the paper devises a rejection sampling method that tries to balance samples in a minibatch. However, a sample with a noisy label can appear to be a "biased" sample (with a correct label) which can affect the proposed method (as well as many other bias mitigation methods). To this end, the paper also proposes a denoising module that successfully eliminates the effects of noisy labels on the debiasing algorithm proposed.

**Summary Of The Review:**

Overall, I think the paper shows lots of properties of a solid paper. It has an interesting hypothesis, a solid experimental section, and some valuable insights with great results. Taken together, I find that it clearly surpasses the bar for acceptance. However, it still leaves a few things to be desired, which prevents me from giving a higher score.

---

> ### Author Response · Authors · 2021-11-14
> **Response to reviewer WQAv (2/2)**
>
> __Q3__) Next, the proposed algorithm falls under the category called variously as "supervised", "known", or "explicit" bias mitigation algorithm. In other words, the algorithm assumes knowledge about which of the factors were biased so that a suitable "bias-only" model can be trained by leveraging only the "bias". Such methods can be unscalable to all but a single, well-defined type of bias. This could be addressed by demonstrating the algorithm's success in more difficult scenarios (https://arxiv.org/abs/2104.00170).
>
> __A3__) We evaluated much more difficult scenarios by additionally testing  ``Biased MNIST,’’ which was developed by the authors of https://arxiv.org/abs/2104.00170. We used the proposed hyperparameters and network architectures from the author of that paper. However, as noted in the official code, the setting in the paper is not yet opened; therefore, we tested under our implementation and reported results, which can differ from the table reported by the authors of Biased MNIST. We compared the proposed method with LfF, which reportedly has the best performance in the Biased MNIST setting.
>
> Biased MNIST 90%
>
> |Alg|Acc|
> |:---:|:---:|
> |LfF		|	__51.45%__|
> |Ours		|	__66.50%__|
>
> RUBi: Reducing Unimodal Biases in Visual Question Answering NeurIPS19
> LearnedMixinH: Don’t Take the Easy Way Out: Ensemble Based Methods for Avoiding Known Dataset Biases EMNLP19
> LfF: Learning from Failure: Training Debiased Classifier from Biased Classifier NeurIPS20
> REPAIR: REPAIR: Removing Representation Bias by Dataset Resampling CVPR19
> Biased MNIST: An Investigation of Critical Issues in Bias Mitigation Techniques WACV21 (https://arxiv.org/abs/2104.00170)

---

> > ### Comment · Reviewer_WQAv · 2021-11-19
> > **Thanks for new experiments**
> >
> > Thank you for your response and new experiments. While there is not a lot of time right now, I would love to see more details about Biased MNIST numbers in the final version (provided that the code/data is fully available). The improvement over LFf seems really promising if it is indeed done on an apples-to-apples basis. I have also read other reviews and the authors' responses to those reviews. While there are some new issues that I did not notice in my original review, I think there weren't enough concerns for me to lower my scores. I tend to keep my original rating of a weak accept.

---

> ### Author Response · Authors · 2021-11-14
> **Response to reviewer WQAv (1/2)**
>
>
> We are grateful for your time and constructive feedback; we appreciate all of your  thoughtful feedback. (1) How to implement a biased model in RUBi, LearnedMixin, REPAIR, and ours. (2) A scalability issue. (3) Results for significantly more challenging tasks. Each of your concerns is addressed individually.
>
> ---
> __Q1__) Unclear how bias model is implemented: In prior works, such as LearnedMixinH and Rubi for VQA, the bias-only model takes question-only model as a biased model which assumes that if an answer could be guessed only using the question, then that sample can be assumed to be biased. There isn't a straightforward parallel for this in color-MNIST. For color-MNIST, REPAIR proposes to train a classifier using only the RGB value of the input. It is unclear how the biased model is trained for the proposed algorithm.
>
> __A1__) Both the bias-only and the question-only models are used as biased models in each publication. Both models are essentially trained on the biased data, "question" in VQA. Note that, in the VQA task, "question" and "image" are clearly separated. Therefore, the biased model can be trained on only the ``question’’ information.
>
> To import both algorithms, we simply change "question" to "color" in the colored MNIST benchmark. However, unlike VQA algorithms, we do not need to explicitly use ``color’’ information when training the biased model. This is because as empirically stated in the LfF paper-the model trained on colored MNIST is biased to color because the color is easier to learn than digit shape target. As a result, we can train the biased model using just colored MNIST itself.
>
> Similarly, in the official REPAIR code constructs a biased model using a linear classifier. As you mentioned, this linear classifier is trained solely on RGB values. However, we strongly believe that it is difficult to assume that bias-related information is provided. Therefore, we employ an image classifier rather than the linear classifier and train it on the provided training set, i.e., colored MNIST. As mentioned previously, this image classifier is color-biased.
>
> As with REPAIR, our biased model is trained on the colored MNIST itself; the only difference is the objective function. As in the LfF paper, we use generalized cross entropy (GCE) loss to intensify bias.
>
> ---
>
> __Q2__) One common theme across these datasets is that they can be "learned" (at least the biased version) with a much smaller amount of data than is present in the training set. Hence, a rejection sampling-based method can work even when the minority set (m) diminishes.
>
> __A2__) No matter whether a given task is difficult or not, we strongly believe that minority samples can also be correctly predicted using a very small amount of minority data. This is because the target attributes are both in majority and minority samples. The main goal of oversampling is to prevent the model from learning bias attributes.
> If our hypothesis is met in the colored MNIST case, the model may correctly predict the sample whose color is not visible in the training set; we empirically checked this using unseen color test images. These images are uniformly randomly colored, i.e., RGB $\sim U(0,1)^3$. These experiments were conducted using models trained on 99.5\% without noisy labels and the test results are listed below.
>
> |Alg|Acc|
> |:---:|:---:|
> |Vanilla	|	50.80%|
> |LfF		|	85.04%|
> |Rebias	|	51.49%|
> |REPAIR	|	53.28%|
> |Ours	|	95.44%|
>
> As stated, the proposed method achieves higher accuracy on the unseen color test.
> Additionally, as described in the Appendix H class activation map (CAM) paragraph, a de-biased model trained on the proposed algorithm learns where to concentrate its attention. In this CAM result, the target model trained based on our algorithm does not see the biased objective in the watermarked MNIST case. We believe that both results support our claims that the proposed method guides the de-biased model not to learn the biased attribute.

---

### Author Response · Authors · 2021-11-14
**Summary of the revision.**

Dear Reviewers and AC,

We would like to express our deepest gratitude for your thoughtful feedback and insightful comments. We have carefully revised our manuscript in response to your question and concerns, including the addition of the following experiments and discussions.

To begin, the incorrect results are contained in Table 2, and thus this section has been corrected. We apologize for any confusion caused. This table is achievable via our initial submission code in supplementary.

Our revisions are summarized in the following items.

- Clearer description about implementation of the prior works for RWQAv-1, WAv-2 (Appendix C)
- Ablation studies on unseen test data for Reviewer WQAv-3 (Appendix H)
- Ablation studies on rarely biased case for Reviewer BR78-3 (Appendix H)
- Clearer description about Table 2 for Reviewer BR78-4 (Section 6.2)
- Histogram of the per-sample gradient statistics for Reviewer 3HUA-1, BR78-2, sFXQ-1 (Section 4.1)
- Comparison with other de-noising algorithm for Reviewer sFXQ-2 (Section 5 and Appendix I)
- Clearer description about our experimental setting for Reviewer sFXQ-3 (Appendix A)

The revisions are marked with "red" in the manuscript.

Best regards,
Authors

---

### Decision · Program_Chairs · 2022-01-20

**Decision:**

Reject

**Comment:**

The manuscript proposes a method to adjust a biased model without requiring explicit annotations of biases. The main hypothesis of the manuscript is that there are differences in the direction and magnitude of the loss gradients for underrepresented samples compared to majority patterns in the training data. Based on this hypothesis, the manuscript proposes a rejection sampling method that tries to balance samples in a minibatch. However, a sample with a noisy label can appear to be an underrepresented sample with a correct label which can affect the proposed method. To tackle this, the manuscript also proposes a denoising module that successfully eliminates the effects of noisy labels on the debiasing algorithm proposed. Experiments are performed on various synthetic and real-world biased sets.

Positive aspects of the manuscript includes:
1. The results for varying levels of "bias" as well as the success of the proposed "denoising" setup is remarkable for the datasets tested;
2. An interesting hypothesis about the differences between gradient magnitude and direction (as measured by its proximity to an "average" gradient direction for all samples) look different for underrepresented samples as compared to "regular" data sample.

There are also several major concerns, including:
1. Lack of motivation and analysis on the connections between per-sample gradients and the majority/minority splits in more complex datasets;
2. The key assumption which motivates the proposed method, namely that minority samples have different gradient distributions than majority ones, deserves a more rigorous validation;
3. The "scalability" of the proposed method. One common theme across these datasets is that they can be "learned" (at least the biased version) with a much smaller amount of data than is present in the training set. Hence, a rejection sampling-based method can work even when the minority set diminishes;
4. Assumption about known bias. The proposed method assumes knowledge about which of the factors were biased so that a suitable "bias-only" model can be trained by leveraging only the "bias".

Post-rebuttal, reviewers stayed with borderline ratings, and they have suggested further improvements: more details about Biased MNIST numbers (to address concerns about known bias), and ablation studies on real datasets (e.g. compare to results without denoising, or denoised using FINE) to fully justify the practical importance of proposed denoising module.